# UnSAF: A Self-Assessment Framework of Uncertainty Awareness for Multimodal LLMs

## Abstract

While Multimodal Large Language Models (MLLMs) demonstrate significant potential in addressing complex multimodal tasks, they often produce plausible yet incorrect responses that limit their practical deployment, highlighting the critical need for reliable uncertainty evaluation. Existing metrics for assessing model uncertainty typically require extensive labeled datasets and rely on token-level confidence, which might be inadequate for open-ended multimodal tasks. To address these issues, we propose an **Un**certainty-Aware **S**elf-**A**ssessment **F**ramework (**UnSAF**), which explicitly incorporates the key question—Do MLLMs know what they don't know?—into the evaluation procedure. Specifically, UnSAF first prompts MLLMs to generate a set of both answerable and unanswerable questions, then requires the models to answer these self-generated questions. The responses are then categorized into four distinct types, namely *true answerable*, *false answerable*, *true unanswerable*, and *false unanswerable*, and this ultimately yields an interpretable and label-free uncertainty-aware F1 (UnF1) score. We conduct extensive studies across both open-source and commercial MLLMs based on UnSAF. Our experiments not only demonstrate the effectiveness of UnSAF compared to conventional metrics but also reveal intriguing observations. Notably, we identify a clear positive correlation between the UnF1 score and model scale, which motivates the use of knowledge distillation to enhance uncertainty awareness in open-source, smaller-scale MLLMs. Unlike simply transferring question-answering ability from larger models, we incorporate uncertainty-aware question generation into the distillation framework by teaching the student model to generate both answerable and unanswerable questions in response to different types of instructions. Experiments show that distilling uncertainty-aware question generation capability markedly enhances MLLMs' uncertainty awareness without degrading original task performance and also noticeably reduces hallucinations.

## 1 Introduction

Multimodal Large Language Models (MLLMs), particularly current large vision-language models (LVLMs), have made significant progress in integrating visual and linguistic modalities, demonstrating strong capabilities in fundamental tasks such as image captioning and visual question answering (VQA). Unfortunately, current MLLMs often produce over- or under-confident predictions (Whitehead et al., 2022) and occasionally hallucinate (Wang et al., 2023; Li et al., 2023b). These systemic deficiencies constrain deployment reliability, as ensuring trustworthy interaction requires models that can understand and express their uncertainty, a capability that remains a substantial challenge in modern MLLMs (Khan & Fu, 2024b; Hartsock & Rasool, 2024; Liu et al., 2025). Conventional uncertainty evaluation metrics, including the Brier score (Glenn et al., 1950) and Expected Calibration Error (ECE) (Naeini et al., 2015), require access to the ground-truth labels and their effectiveness deteriorates when data are scarce (Kuhn et al., 2023; Kumar et al., 2019). Moreover, these metrics are designed for closed-set settings with predefined label sets and are therefore inadequate for open-ended multimodal tasks (Chen et al., 2025). In such settings, semantically equivalent outputs, such as *a man is riding a bike* and *a person is on a bicycle*, remain difficult to evaluate with token-level exact string matching. While approaches such as semantic entropy (Kuhn et al., 2023) have been proposed to mitigate this issue, the

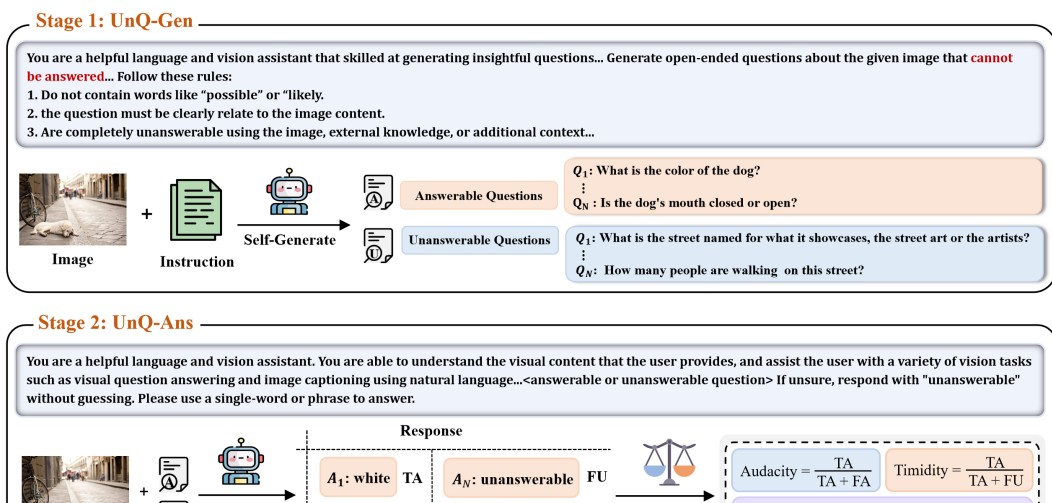

Figure 1: Overview of UnSAF for uncertainty evaluation of MLLMs. The model first generates questions it deems answerable and unanswerable, and then the model is required to answer these questions. The answer is naturally categorized into four types, which jointly define the UnF1 score.

continuous nature of semantic similarity makes entropy estimates highly sensitive to the choice of clustering threshold (Farquhar et al., 2024), thereby reducing robustness and reproducibility. Beyond this, recent uncertainty evaluation approaches for MLLMs also face limitations. Misleading Rate (MR) (Dang et al., 2025) measures the proportion of answer flips under misleading prompts. While this effectively reveals the model's vulnerability to adversarial inputs, it remains inapplicable to open-ended generation. Consistency and Uncertainty (CU) (Khan & Fu, 2024a) relies on rephrasing, but it suffers from metric instability in open-ended tasks due to the inherent difficulty of accurately equating free-form answers. These limitations highlight a critical gap in the field and necessitate a more generalized uncertainty evaluation method.

To address these limitations, we propose a novel uncertainty evaluation framework for MLLMs, termed Uncertainty-Aware Self-Assessment Framework (UnSAF). As illustrated in Figure 1, the framework operates in two stages: (1) the model generates questions that it deems to be answerable and unanswerable; (2) the model is required to answer this self-generated set of questions, with each of its response categorized into one of four types: True Answerable (TA), False Answerable (FA), True Unanswerable (TU), and False Unanswerble (FU). Leveraging this formulation, we can calculate the Uncertainty-aware F1 (UnF1) score, an interpretable and label-free metric to evaluate uncertainty awareness of MLLM. In contrast to conventional metrics, UnSAF evaluates uncertainty awareness without relying on enormous labeled samples and remains effective for open-ended tasks. We comprehensively evaluate UnSAF across 14 open-source and three commercial MLLMs, validating its effectiveness and revealing several intriguing observations. Our validity analysis demonstrates that the UnF1 score yields a consistent ranking across different datasets and remains stable under sampling variations, whereas conventional metrics exhibit high volatility, especially in low-data regimes. The empirical analysis based on UnSAF reveals that instruction-tuned MLLMs tend to adopt more timid behavior. Although prompt engineering can partially mitigate this issue, increased abstention lowers the rate of false answers at the cost of more false unanswered cases, thereby limiting the model's expressive capacity. Furthermore, our experiments based on UnSAF reveal a clear positive correlation between model size and UnF1 score, suggesting that larger models tend to exhibit improved uncertainty awareness.

Motivated by the observed scaling effect, we propose a distillation framework to enhance uncertainty awareness in open-source, smaller-scale MLLMs. Following UnSAF's two-stage pipeline, we construct uncertainty-aware datasets with annotations of instruction–question–answer triplets curated with GPT-4o and peer MLLMs. We propose three distillation variants—UnD-QA, UnD-IQ, and UnD-Joint—trained on images with question–answer pairs, instruction–question pairs, and their union, respectively. These variants target distinct facets of uncertainty awareness, and all student models are LoRA-finetuned on the distilled data to align with these uncertainty-aware dialogues.

We believe that prior attempts to train models to abstain on unanswerable questions (Miyai et al., 2024; Guo et al., 2024) have produced only marginal improvements, largely because they rely solely on fine-tuning with QA pairs. In contrast, our approach integrates uncertainty-aware question generation into distillation, which we find to be the most critical for imparting uncertainty awareness, achieving an average 15% improvement in UnF1 score compared to the inconsistent 2% from QA-based distillation. Importantly, the improvements are consistent, preserve VQA performance, and reduce hallucination, demonstrating that they come without adverse trade-offs.

Overall, our contributions are as follows:

(1) We introduce UnSAF, an interpretable and label-free assessment framework for uncertainty evaluation in both black-box and white-box settings. The resulting UnF1 score retains stable rankings across benchmarks and remains unaffected by sample-size perturbations.

(2) Our empirical investigations with UnSAF reveal that instruction-tuned MLLMs consistently exhibit timid response patterns. However, simply applying prompt engineering proves insufficient to improve the UnF1 score, since it often exacerbates audacity when mitigating timidity, and vice versa. Furthermore, we identify a consistent positive correlation between model scale and UnF1 score performance.

(3) We construct the UnSAF dataset with GPT-4o and design three distillation strategies: UnD-QA, UnD-IQ, and UnD-Joint. We identify that distilling uncertainty-aware question generation yields the largest improvement in UnF1 score without degrading task performance.

## 2 RELATED WORK

### 2.1 UNCERTAINTY EVALUATION

Uncertainty evaluation has been extensively studied in conventional classification tasks, typically relying on output confidence such as softmax probability (Hendrycks & Gimpel, 2017; Guo et al., 2017; Wang et al., 2021) or predictive entropy (Gal & Ghahramani, 2016). Common metrics include proper scoring rules such as the Brier score (Glenn et al., 1950) and its variants (e.g., MacroCE (Si et al., 2022)), as well as binning-based calibration metrics such as ECE and Maximum Calibration Error (MCE) (Naeini et al., 2015). In the context of (multimodal) LLMs, these metrics have been directly adapted (Chen et al., 2025; Yang et al., 2023; Lau et al., 2025), which particularly focus on multi-choice settings without considering the unique characteristics of generative tasks, in which models generate free-form responses rather than choose from predefined label sets (Chen et al., 2025). Although existing methods like semantic entropy (Kuhn et al., 2023) attempt to evaluate uncertainty in free-form generation, there remains a significant need for efficient and robust uncertainty evaluation frameworks in open-ended settings.

### 2.2 SELECTIVE PREDICTION AND ABSTENTION BEHAVIOR

Selective prediction aims to enable models to withhold answers when uncertain, thereby reducing the risk of producing incorrect outputs (Geifman & El-Yaniv, 2019). Conventionally, selective prediction has been studied in unimodal, closed-world tasks such as image classification (Geifman & El-Yaniv, 2017), and has only recently been extended to multimodal, open-ended tasks like VQA (Whitehead et al., 2022; Dancette et al., 2023). In this context, Whitehead et al. (2022) proposed to train a selection function using features from held-out validation data, while Dancette et al. (2023) proposed to promote abstention by training models on curated uncertainty-aware data slices. More recent approaches leverage answer self-consistency signals to decide whether to respond or abstain (Khan & Fu, 2024b) or enhance the diversity of selection functions (Guo et al., 2024). These works mainly focus on reducing the risk of incorrect answers, while our work integrates the abstention behavior directly into the generative dynamics of MLLMs.

### 2.3 INSTRUCTION TUNING FOR MLLMS

Instruction tuning has become essential for aligning language models with human intent across diverse tasks (Ouyang et al., 2022). In multimodal settings, this paradigm extends to vision-language instruction tuning (VLIT), where models are trained to follow instructions grounded in

both textual and visual inputs (Liu et al., 2023; Li et al., 2024; 2023a; Chen et al., 2023). The efforts towards forming such a dataset into the abstention ability of multi-modal large models are only the beginning (Miyai et al., 2024). However, using this kind of dataset to supervised finetuning is not always functional (Kapoor et al., 2024). Unlike just forming a dataset or improving uncertainty awareness via supervised fine-tuning QA pairs (Chandu et al., 2024; Guo et al., 2024), we also propose supervised fine-tuning using instruction-question pairs instead of solely on QA pairs. This design explicitly targets uncertainty awareness in MLLMs, representing, to the best of our knowledge, a novel contribution.

## 3 UnSAF Strategy

### 3.1 Conventional metrics for model uncertainty evaluation

Given that MLLMs inherently generate token-level probability distributions, we first consider conventional uncertainty evaluation metrics for classification.

**Conventional Metrics.** The Brier score (Glenn et al., 1950) and its variant MacroCE (Si et al., 2022) capture the deviation between predicted probabilities and ground-truth outcomes, while ECE and MCE (Naeini et al., 2015) assess the alignment between model confidence and empirical accuracy (Further details are provided in Appendix B). However, these metrics rest on assumptions underlying conventional classification and encounter fundamental limitations when applied to MLLMs.

**Multi-choice and Open-ended Setting.** Although the multi-choice format superficially resembles classification, generative prompts will emit a non-option prefix as the first token, potentially leading the model to misinterpret the probability of the prefix as that of the full target answer, e.g., *ans* in *answer is C*. The next-token distribution reflects the likelihood of generating such prefixes rather than the model's true preference over candidate answers, and consequently, confidence values derived from token probabilities are unreliable. In open-ended generation, the situation is even more challenging, arising from two main issues: (1) Tokenization artifacts (e.g., spaces, punctuation) that can distort probability estimates, and (2) the fact that correct answers may appear in multiple valid surface forms (e.g., synonyms, casing, formatting) makes exact string matching overly restrictive. The limitations reveal a fundamental disconnect between conventional metrics and the ways in which MLLMs express uncertainty in real-world interactions. To address this gap, we propose the Uncertainty-aware Self-Assessment Framework (UnSAF). Unlike the Brier Score and ECE measure probability calibration, UnSAF evaluates answer and abstention consistency.

### 3.2 Uncertainty-aware Self-Assessment Framework

**Uncertainty-aware Question Generation.** As shown in Fig. 1, in the initial stage UnQ-Gen, MLLMs are prompted to generate both answerable and unanswerable questions grounded in the given images under diverse instructions. In the multi-choice setting, the model generates questions each accompanied by four predefined options ($A$, $B$, $C$, $D$) that are manually verified to contain exactly one unambiguously correct answer. In the open-ended setting, the model is instead asked to produce free-form responses that can range from a short phrase to several sentences. For either format, the model is not tasked with providing an answer, as the subsequent evaluation stage assesses only the binary [answerable/unanswerable] label assigned to each generated question. In fact, the ability to produce a question that corresponds to the instructed binary label is itself a measure of the model's uncertainty awareness, thus, when the correctness of this label is not our concern. Additionally, an analysis of the sensitivity to the ratio of answerable versus unanswerable questions is provided in Appendix E.5.

**Uncertainty-aware Question Answering.** The second stage UnQ-Ans poses each self-generated question in a new conversational session. The model is instructed to output unanswerable whenever it is uncertain. In the multi-choice setting, the model selects one of four options when confident, whereas in the open-ended setting, it generates free-form responses that can range from a short phrase to several sentences; in both cases, it outputs unanswerable otherwise. Since this step is conducted in a new session, models with poorer uncertainty awareness tend to exhibit poor alignment with the collected binary labels from the first stage, regardless of the quality of the questions generated in that stage. Prompt templates for the two stages are provided in Appendix J. To further

validate the robustness of our framework against prompt variations, we present a detailed sensitivity analysis in Appendix E.8, which confirm that UnSAF is not susceptible to prompt influence.

Within the UnSAF framework, the model's behavior is assessed based on whether it provides an answer or abstains from answering these self-generated questions, leading to four categories: (1) True Answerable (TA), means the model answers the answerable question; (2) False Answerable (FA), means the model answers the unanswerable question; (3) True Unanswerable (TU), means the model abstains from answering the unanswerable question; (4) False Unanswerable (FU), means the model abstains from answering the answerable question. From these counts, we derive audacity, timidity, and their harmonic mean, the uncertainty-aware F1 (UnF1) score as follows:

$$Audacity = \frac{\text{TA}}{\text{TA} + \text{FA}}, \; Timidity = \frac{\text{TA}}{\text{TA} + \text{FU}}, \; UnF1 = 2 \cdot \frac{Audacity \cdot Timidity}{Audacity + Timidity}. \quad (1)$$

Here, audacity and timidity are structurally analogous to precision and recall in binary classification. Audacity is defined as the proportion of answerable questions among all answered questions, where a lower value indicates a tendency to answer aggressively. Conversely, timidity is defined as the proportion of answered questions among all answerable questions, where a lower value reflects a tendency to abstain more frequently. Finally, the UnF1 score captures the model's ability to decide judiciously when to answer and when to abstain. Specifically, when a model abstains from answering all self-generated questions, resulting in $TA = FA = 0$, and consequently the audacity becomes mathematically undefined due to division by zero. In such instances, we explicitly define the UnF1 score as 0.

## 4 COMPREHENSIVE EVALUATION

In this section, we empirically study UnSAF in terms of its validity and experimental findings. Section 4.1 examines whether it exhibits consistency across datasets and varying sample sizes, and Section 4.2 investigates the impacts of model scaling, instruction tuning, and prompt engineering.

**Models.** We benchmark five open-source MLLM families to capture diversity in model scale and training corpora: LLaVA-1.5-7B/13B, Qwen2-VL-7B, Qwen-VL-Chat-7B, Qwen2.5-VL-Instruct-7B/32B, InternVL2-8B/14B, mPLUG-Owl-2B/7B, and MiniCPM-3B/8B. [1] In addition, we include three commercial MLLMs: GPT-4o, Moonshot-v1, and GLM-4V-Plus. All experiments use customized, family-specific prompt templates (detailed in Appendix J.3), 16-bit precision, greedy decoding, and no task-specific fine-tuning; each result averages three random seeds unless otherwise noted. All runs are conducted on NVIDIA H100 (80 GB) GPUs.

### 4.1 VALIDITY ANALYSIS OF UNSAF

Conventional metrics such as Brier score and ECE are widely used for evaluating uncertainty (Tu et al., 2024; Chen et al., 2025), yet they often violate two practical requirements: (i) consistent model rankings across datasets and (ii) robustness under varying sample sizes. Given these shortcomings, we first compare the consistency of the UnF1 score with that of Brier score, MacroCE, ECE, MCE, MR and CU across diverse datasets, and examine its stability relative to ECE, MR and CU under varying sample sizes.

**UnSAF preserves model rankings across datasets.** We compare all metrics with the exception of MR across five benchmarks: one multiple-choice dataset: MMBench, and four open-ended datasets: VQAv2, VizWiz, TextVQA, and OKVQA, covering diverse visual domains and question types. Given that MR is applicable only to multi-choice tasks, we evaluate it on a separate set of five datasets: ConBench, MMBench, MME, MMMU, and ScienceQA (corresponding to the rankings in Fig. 2). From each dataset, 4k instances are randomly sampled, and five representative MLLMs are evaluated. ECE and MCE are computed with $M=15$ confidence bins. We rank five MLLMs within each dataset according to their scores under each uncertainty metric and assess their stability across datasets using Spearman's Footrule (SF) distance (Appendix C), where lower values indicate more consistent rankings.

---

[1] Not every experiment involves the entire model set; the specific subset is stated in each study. Unless otherwise noted, we report results for the variant closest to the 7B scale.

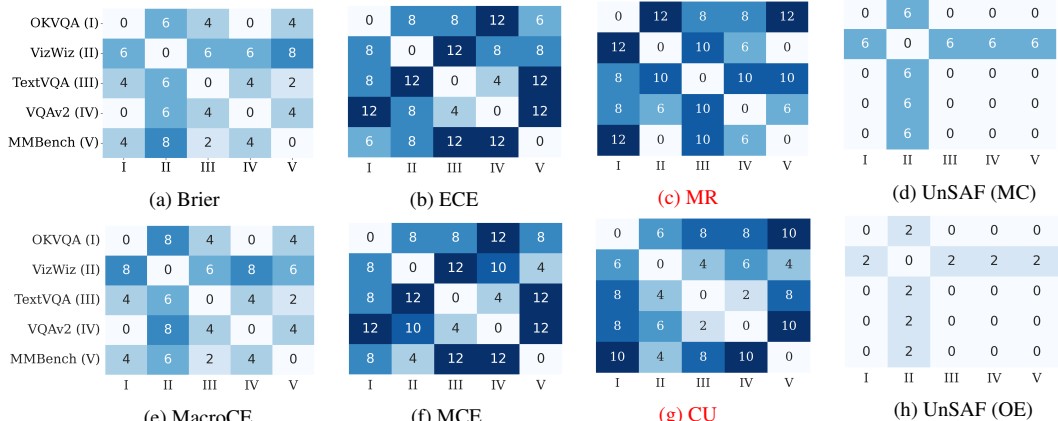

Figure 2: Spearman's Footrule distances for model rankings across datasets, revealing high instability for conventional metrics while UnF1 score remains essentially invariant.

Figure 2 illustrates SF distances across datasets, highlighting the comparative stability of five uncertainty metrics. Brier score yields moderate instability, with distances ranging from 0 to 8 and mean values around 4.2, indicating frequent rank reversals. MacroCE is comparatively more stable, with most distances below 4 and a mean near 2. In contrast, both ECE and MCE exhibit severe instability, with distances frequently reaching the maximum reversal 12 and mean values around 9. The MR metric exhibits pronounced ranking divergence, with distances concentrated between 6 and 12, while CU shows comparable instability, spanning a broad range of 2 to 10, reflecting frequent rank reversals across datasets. Strikingly, the UnF1 score produces nearly invariant rankings across benchmarks: in both the multiple-choice and open-ended settings, the distances are consistently zero, with only minor deviations observed on VizWiz (ranging from 2 to 6). These results highlight the superiority of the UnF1 score to maintain consistent rankings across diverse datasets, establishing it as a reliable metric for multimodal uncertainty evaluation in the wild.

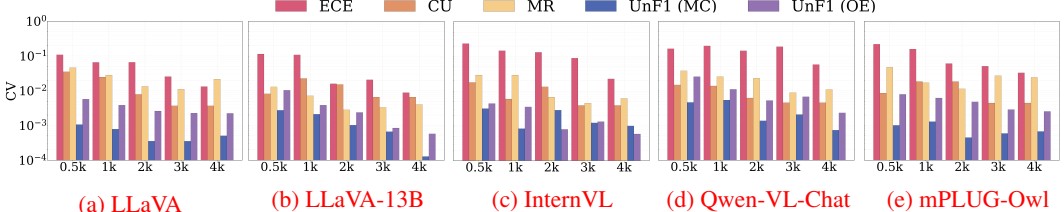

Figure 3: Coefficient of variation comparison between UnF1 score and ECE across varying sample sizes for five models. UnF1 remains consistently low and stable across both multi-choice and open-ended settings while ECE, CU, MR exhibits substantially higher CV, particularly at smaller $n$.

**UnSAF demonstrates robustness to varying dataset sizes.** Each metric's robustness to dataset size is examined on the same five MLLMs. For each run with a specific dataset size $n$, we report results of ECE, MR the UnF1 score averaged over five independently resampled subsets of MMBench while MR over OKVQA. Since they are differ in scale, we compare their coefficient of variation (CV), defined as CV $= \sigma/\mu$, where $\sigma$ denotes the standard deviation across the resampled subsets and $\mu$ denotes their mean.

Figure 3 shows the CV of ECE, CU, MR and the UnF1 score across sample sizes. ECE exhibits substantially higher CV, one to two orders of magnitude above that of UnF1 score, fluctuating markedly at small $n$ and stabilizing only partially as $n$ grows. CU and MR, despite outperforming ECE, fail to achieve convergence, maintaining distinctively higher instability levels compared to the UnF1 score. In contrast, the UnF1 score exhibits consistently low CV values that decrease with $n$, approaching zero for all models. Notably, UnF1 score exhibits almost identical curves under both the multiple-choice and open-ended settings, highlighting the robustness of UnSAF. Overall, these results demonstrate that UnSAF offers greater robustness to variations in dataset size compared with ECE. Detailed ECE mean and standard deviation results for each model are provided in Appendix H.

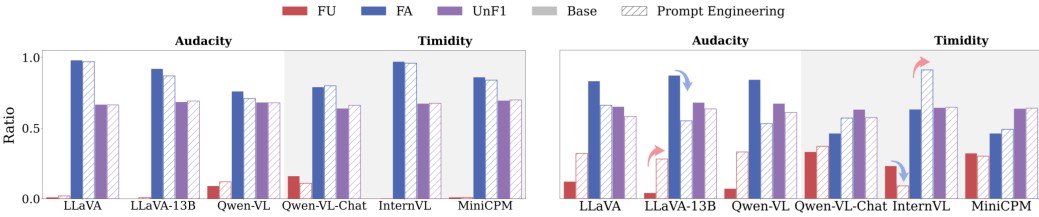

Figure 5: Effect of prompt engineering on audacity, timidity, and UnF1 score under multi-choice (left) and open-ended (right) settings. Prompt engineering reduces FA in audacity models but increases FU, whereas timidity models (distinguished in the gray area) exhibit the opposite trade-off.

## 4.2 FINDINGS BASED ON UNSAF

Having validated UnSAF, this section analyzes MLLMs' uncertainty using the proposed UnF1 score, focusing on the effects of instruction tuning, prompt engineering, and model size scaling.

**Instruction-Tuned models are more timid.** Instruction tuning can alter model behavior in ways that diverge from its base counterpart. To investigate this effect, we compare base and instruction-tuned variants for open-ended VQA tasks. As shown in Figure 4, instruction tuning induces a pronounced reduction in timidity, reflecting a higher tendency of the tuned models to abstain. Audacity remains largely unchanged, indicating that the dominant behavioral shift is increased abstention rather than heightened overconfidence. This imbalance causes a marked drop in UnF1 across multiple datasets. This aligns with prior evidence that pre-trained models tend to be well-calibrated, yet post-

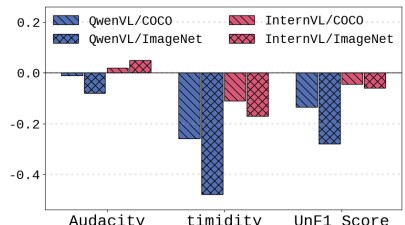

Figure 4: Performance differences between instruction-tuned models and their pretrained counterparts.

training via reinforcement learning frequently degrades their uncertainty calibration. We also refer interested readers to Appendix E.3, where we further report the absolute values of audacity, timidity, and UnF1 score for both the base and instruction-tuned models.

**Prompt engineering provides limited uncertainty-aware benefits.** We quantify each model's response disposition along the timidity–audacity axis by comparing FU and FA rates: a higher FU indicates greater abstention, whereas a more elevated FA indicates overconfident answering on unanswerable questions. As Figure 5 shows, under the multiple-choice format, every model consistently selects one of the four options, yielding uniformly high audacity irrespective of its scale or architecture. By contrast, in the open-ended setting, LLaVA-7B/13B and Qwen-VL-7B exhibit audacious behavior with higher FA and lower FU, while Qwen-VL-Chat-7B, InternVL-8B, and MiniCPM-8B show timidity with relatively lower FA and higher FU. To address these tendencies, we explore prompt engineering as a lightweight intervention: encouraging timid models to answer more confidently and guiding audacious models to respond more conservatively (the full prompt templates are provided in Appendix J.4). Figure 5 shows the changes in FU, FA, and UnF1 score before and after applying prompt engineering under both multiple-choice and open-ended settings. The effect is limited in the multiple-choice setting, whereas in the open-ended setting, prompts reduce FA in audacious models but increase FU, leading to more false abstentions. Symmetrically, timidity models experience a reduction in FU at the expense of higher FA. Consequently, prompt engineering alone fails to achieve a satisfactory trade-off between audacity and timidity, leaving overall gains in uncertainty awareness inconsistent and limited. Please refer to Table 6 in Appendix E.4 for the detailed numerical results.

**Impact of model size on uncertainty awareness mirrors accuracy.** We next examine how model scale and type influence uncertainty awareness. Figure 6 demonstrates that the evolution of uncertainty awareness evolves with scale in close parallel to predictive accuracy: larger MLLMs deliver superior task performance and markedly higher UnF1 score, underscoring a robust positive correlation between accuracy and uncertainty awareness. Beyond this general trend, Figure 6 also reveals a sharp, almost discontinuous rise in uncertainty awareness once the model reaches a

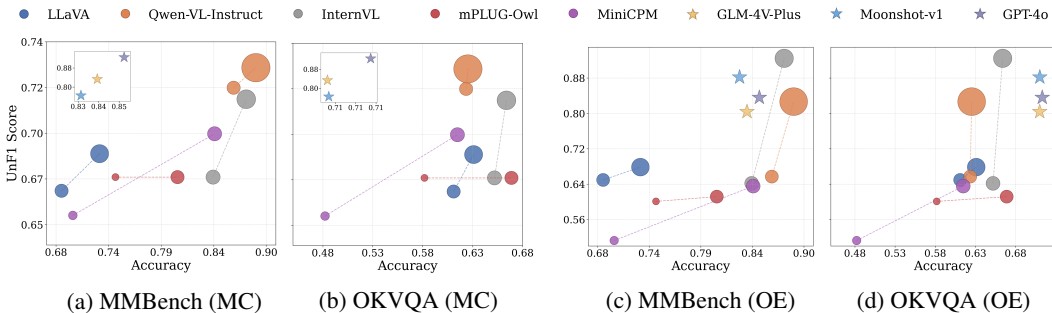

Figure 6: Bubble Graph based on the metrics of accuracy (x-axis), UnF1 score (y-axis), and parameter scale (bubble size). Larger models yield higher accuracy and higher UnF1 score.

critical scale. For MMBench under the open-ended setting, UnF1 score rises steeply once accuracy surpasses approximately 80%; further parameter growth contributes marginal gains in accuracy yet yields disproportionately large improvements in uncertainty awareness. OKVQA follows an analogous trajectory, though with a lower inflection point near 64%. In both settings, these near mirror-symmetric L-shaped curves persist, indicating that further parameter growth contributes marginal gains in accuracy yet yields disproportionately large improvements in uncertainty awareness.

## 5 UNCERTAINTY-AWARE DISTILLATION

Motivated by the scaling effect in uncertainty awareness observed above, in this section, we leverage Uncertainty-aware Distillation to improve open-source, smaller-scale MLLMs. Section 5.1 describes the distillation dataset construction and three distillation strategies, and Section 5.2 provides a detailed evaluation of their impact.

### 5.1 DISTILLATION DATASET AND STRATEGIES

**Dataset Construction.** To enact uncertainty-aware distillation, we first construct a distillation dataset by replicating UnSAF's two-stage pipeline. The objective is to transfer the teacher MLLMs' superior ability to discriminate between answerable and unanswerable queries into the student model. Concretely, we randomly draw 1k images from the COCO training split; for each image, the teacher first generates five answerable and five unanswerable questions via UnQ-Gen, and then independently answers every question following UnQ-Ans, yielding 10k images and the corresponding instruction–question pairs and an equal number of question–answer pairs. The complete distillation datasets, generated by GPT-4o and Moonshot-v1, will be released to facilitate research on uncertainty-aware distillation for open-source MLLMs.

**Distillation Strategies.** To examine the contribution of different dataset components to uncertainty awareness, we consider three complementary strategies: (1) **UnD-QA** is a strategy that distills from **Q**uestion-**A**nswer pairs. In this strategy, the student model is trained to replicate the teacher's answering behavior using the teacher's responses. This method of supervised fine-tuning on QA pairs is also employed in prior research (Chandu et al., 2024; Guo et al., 2024). (2) **UnD-IQ** focuses on distilling the ability of questioning rather than answering. Using **I**nstruction-**Q**uestion pairs, the student model learns to generate questions that are deliberately designed to be answerable or unanswerable based on the prompt. Inspired by Voltaire's quote—*Judge a man by his questions rather than his answers*, we investigate whether teaching the students to imitate the question-generation behavior under specific prompt instruction is a better way to improve students' uncertainty awareness. (3) **UnD-Joint** combines the two above. This joint distillation process enables the student to learn both uncertainty-aware questioning and answering simultaneously.

### 5.2 EMPIRICAL EVALUATION OF THE DISTILLATION STRATEGY

**Baselines and setup.** We compare our method with UNK-VQA (Guo et al., 2024) and MM-UPD (Miyai et al., 2024) that aim at improving model uncertainty or abstention capabilities. UNK-VQA explicitly focuses on identifying questions that a model does not know. It features two distinct

Table 1: Performance of student models under different distillation strategies. The best results are shown in bold, and the second-best results are underlined. A and T refer to audacity and timidity.

| Teacher Model | | GPT-4o | | | Moonshot-v1 | | | InternVL-14B | | |
|---|---|---|---|---|---|---|---|---|---|---|
| Student Model | Strategy | A | T | UnF1 | A | T | UnF1 | A | T | UnF1 |
| InternVL-8B | Baseline | 0.65 | 0.90 | 0.76 | 0.65 | 0.90 | 0.76 | 0.65 | **0.90** | 0.76 |
| | MM-UPD | 0.65 | 0.77 | 0.71 | 0.60 | 0.77 | 0.71 | 0.60 | 0.77 | 0.71 |
| | UNK-VQA(OE) | 0.53 | 0.98 | 0.69 | 0.53 | 0.98 | 0.69 | 0.53 | 0.98 | 0.69 |
| | UnD-QA | 0.60 | 0.64 | 0.62 | 0.63 | 0.74 | 0.68 | 0.67 | 0.73 | 0.70 |
| | UnD-IQ | 0.90 | 0.92 | 0.91 | **0.96** | **0.97** | **0.96** | **0.98** | 0.83 | **0.90** |
| | UnD-Joint | **0.91** | **0.96** | **0.93** | **0.96** | 0.95 | 0.95 | 0.74 | 0.86 | 0.83 |
| Qwen-VL-Chat-7B | Baseline | 0.51 | 0.67 | 0.58 | 0.51 | 0.67 | 0.58 | 0.51 | 0.67 | 0.58 |
| | MM-UPD | 0.66 | 0.55 | 0.60 | 0.66 | 0.55 | 0.60 | 0.66 | 0.55 | 0.60 |
| | UnD-QA | 0.58 | 0.49 | 0.53 | 0.55 | 0.69 | 0.61 | 0.53 | 0.63 | 0.58 |
| | UnD-IQ | 0.70 | **0.70** | 0.70 | **0.72** | 0.64 | 0.68 | 0.82 | 0.65 | 0.72 |
| | UnD-Joint | **0.88** | 0.67 | **0.76** | 0.60 | **0.80** | **0.71** | 0.74 | **0.86** | **0.79** |
| mPLUG-Owl-7B | Baseline | 0.59 | 0.63 | 0.61 | 0.59 | 0.63 | 0.61 | 0.59 | 0.63 | 0.61 |
| | MM-UPD | 0.55 | 0.66 | 0.60 | 0.55 | 0.66 | 0.60 | 0.55 | 0.66 | 0.60 |
| | UNK-VQA(BY) | 0.76 | 0.39 | 0.52 | 0.76 | 0.39 | 0.52 | 0.76 | 0.39 | 0.52 |
| | UNK-VQA(OE) | 0.53 | 0.53 | 0.53 | 0.53 | 0.53 | 0.53 | 0.53 | 0.53 | 0.53 |
| | UnD-QA | 0.63 | 0.45 | 0.52 | 0.56 | 0.68 | 0.61 | 0.60 | 0.51 | 0.55 |
| | UnD-IQ | 0.88 | 0.64 | 0.74 | **0.93** | 0.79 | 0.85 | **0.66** | 0.88 | **0.75** |
| | UnD-Joint | **0.93** | **0.97** | **0.95** | 0.93 | **0.91** | **0.92** | 0.62 | **0.90** | 0.73 |

FU  TA  TU  FA

| | | |
|---|---|---|
| (a) InternVL | (b) Qwen-VL-Chat | (c) mPLUG-Owl |

Figure 7: Performance with different distillation strategies when using GPT-4o as teacher model.

settings: a binary setting that predicts *answerable* or *unanswerable* and an open-ended setting that provides the actual answer. MM-UPD is designed to evaluate an MLLM's ability to abstain when facing unanswerable or under-specified visual questions.

We employ two frontier commercial MLLMs GPT-4o, Moonshot-v1 and an open-source MLLM InternVL3-14B as teacher models, because they achieve the superior UnF1 score among evaluated models (Figure 6). The student models are InternVL2-8B, Qwen-VL-Chat-7B, and mPLUG-Owl-7B. All students are fine-tuned with LoRA for 5 epochs using AdamW and a cosine learning-rate schedule. The results of uncertainty awareness are measured on the COCO-test dataset using the UnF1 score. Specifically, the UnD-QA prompts to GPT-4o and Moonshot-v1 employ chain-of-thought–augmented answers, whereas InternVL-14B supplies standard QA pairs, reflecting each model's differing capacity for generating extended reasoning. To demonstrate the generalization of our strategy, additional results on diverse benchmarks are provided in Appendix E.6.

Table 1 summarizes Audacity, Timidity, and UnF1 score for each student model distilled from three teachers, and Figure 10 illustrates the answer distributions using these strategies when using GPT-4o as teacher model, with corresponding results using Moonshot-v1 and InternVL-14B reported in Appendix I. It is observed that both UnD-IQ and UnD-Joint distillation strategies lead to substantial improvements in models' uncertainty awareness performance. Notably, UnD-IQ outperforms UnD-QA by delivering larger and more consistent UnF1 score gains across all student models and teacher combinations. Depending on the architecture, the gains range from 12% to 17%, and similar relative improvements are observed across all three teacher models. UnD-Joint also achieves strong performance, often ranking first or second, by further shifting mass from FA to TA on answerable items. In contrast, UnD-QA offers little or inconsistent benefit, with UnF1 score remaining close to baseline or even declining in some cases. As shown in Figure 10, the gains from UnD-IQ arise from a higher proportion of TU predictions on the unanswerable subset and a corresponding reduction in FA. UnD-Joint further shifts FA toward TA on answerable questions, yielding extra gains by jointly reinforcing both question generation and correct answering. In summary, the results indicate that

| Model | Metric | Method | | | | | |
|---|---|---|---|---|---|---|---|
| | | Baseline | DoLa | CoVe | UnD-QA | UnD-IQ | UnD-Joint |
| InternVL-8B | POPE (*F1* ↑) | 83.98 | **87.70** | 87.10 | 86.01 | 83.63 | 87.11 |
| | CHAIR$_i$ ↓ | 10.21 | 10.02 | 9.67 | **9.60** | 10.89 | 9.72 |
| | CHAIR$_i$ (*F1* ↑) | 73.70 | 73.88 | 73.81 | 73.40 | 74.00 | **74.10** |
| Qwen-VL-Chat | POPE (*F1* ↑) | 86.19 | 79.00 | 78.30 | 84.19 | **90.68** | 90.43 |
| | CHAIR$_i$ ↓ | 8.49 | **8.48** | 8.56 | 9.87 | 9.50 | 9.90 |
| | CHAIR$_i$ (*F1* ↑) | 73.60 | 77.43 | 77.13 | 74.50 | **77.90** | 77.00 |
| mPLUG-Owl-7B | POPE (*F1* ↑) | 86.34 | 79.11 | 79.00 | 91.36 | 88.01 | **91.60** |
| | CHAIR$_i$ ↓ | 11.14 | 10.02 | 7.07 | **6.95** | 10.31 | 9.08 |
| | CHAIR$_i$ (*F1* ↑) | 72.30 | 73.88 | 72.83 | 75.10 | **78.30** | 76.50 |

Table 2: Comparison of hallucination mitigation performance. Our UnD methods (UnD-IQ/Joint) consistently achieve superior performance on POPE and CHAIR metrics across different models compared to inference-time baselines like DoLa and CoVe.

**training models to distinguish answerable and unanswerable questions, rather than simply teaching them to abstain, leads to better uncertainty awareness**.

Regarding the omitted results in table, we excluded them due to two distinct issues. For UNK-VQA (OE) on Qwen-VL-Chat-7B and UNK-VQA (BY) on InternVL-8B, the tuned models partially lost the ability to generate valid questions. Meanwhile, the zero UnF1 score observed with UNK-VQA (BY) on Qwen-VL-Chat-7B indicate that the model defaulted to unanswerable outputs for nearly all self-generated questions. Furthermore, to provide a comprehensive assessment of model performance alongside our UnF1 score, we report both the results of ECE and answer accuracy of distilled models in Appendix E.7.

**Hallucination mitigation.** We also assess whether the distillation reduces hallucinations. Hallucination is assessed following two widely-used benchmarks: POPE (Li et al., 2023b) for object-level detection and the CHAIR framework (Rohrbach et al., 2018) for caption-level hallucination. To test the effectiveness of our method, we compare against two representative inference-time hallucination mitigation baselines: DoLA (Chuang et al., 2024), which employs contrastive decoding across layers, and CoVe (Dhuliawala et al., 2024), which utilizes a multi-step self-verification process.

As summarized in Table 2, all three distillation variants under teacher InternVL-14B consistently lower hallucination rates across metrics, clearly showing that the proposed distillation paradigm reliably enhances the grounding of generated content and curbs spurious object mentions. It is worth noting that while DoLa and CoVe require complex inference-time interventions, our method achieves comparable or superior results via standard inference, making it more efficient for deployment. Furthermore, complex inference-time interventions may harm question-answering ability, as POPE scores are lower than the baseline, while CHAIR scores are not. More detailed numerical results of accuracy and hallucination, and details about the hallucination metrics are provided in Appendix F and G.

## 6 CONCLUSION

We introduce UnSAF, a self-assessment framework that enables label-free evaluation of uncertainty awareness in MLLMs while overcoming critical limitations of existing metrics. Extensive experiments show the validity and robustness of UnSAF across diverse datasets and varying sample sizes. Leveraging UnSAF, we conduct empirical studies on MLLMs of different scales and types, revealing several intriguing findings. Building on these insights, we propose an uncertainty-aware distillation framework with multiple strategies. Notably, the distillation experiments show that distilling the teacher's ability to generate uncertainty-aware questions yields substantial improvements in uncertainty awareness without compromising task performance.

REPRODUCIBILITY STATEMENT

For the reproducibility of our results, we provide a detailed description of our methods and experimental setups in Section 4 and Section 5.2. To further facilitate reproduction, table 3 lists all datasets, models, and frameworks used in our experiments, together with their public links or endpoints. **The source code package has been uploaded to the OpenReview system as supplementary material, and we will release all the source codes, fine-tuned model checkpoints, and the distillation datasets on the project page.**

Table 3: Details for reproducibility.

| Name | Public Link or Endpoint |
|---|---|
| **Datasets** | |
| COCO | https://cocodataset.org/#download |
| ImageNet | https://image-net.org/ |
| MMBench | https://huggingface.co/datasets/lmms-lab/MMBench/viewer/en |
| VizWiz | https://huggingface.co/datasets/lmms-lab/VizWiz-VQA/viewer/default/val |
| VQAv2 | https://huggingface.co/datasets/lmms-lab/VQAv2 |
| OK-VQA | https://huggingface.co/datasets/lmms-lab/OK-VQA |
| TextVQA | https://huggingface.co/datasets/lmms-lab/textvqa |
| ConBench | https://huggingface.co/datasets/ConBench/ConBench |
| MMME | https://huggingface.co/datasets/lmms-lab/MME |
| MMMU | https://huggingface.co/datasets/MMMU/MMMU |
| ScienceQA | https://huggingface.co/datasets/derek-thomas/ScienceQA |
| **Models for Evaluation** | |
| GPT-4o | gpt-4-turbo-2024-04-09 |
| Moonshot-v1 | moonshot-v1-8k-vision-preview |
| GLM-4V-Plus | GLM-4V-Plus-0111 |
| LLaVA-v1.5-7B | https://huggingface.co/llava-hf/llava-1.5-7b-hf |
| LLaVA-v1.5-13B | https://huggingface.co/llava-hf/llava-1.5-13b-hf |
| Qwen2-VL-7B | https://huggingface.co/Qwen/Qwen2-VL-7B |
| Qwen-VL-Chat-7B | https://huggingface.co/Qwen/Qwen-VL-Chat |
| Qwen2.5-VL-Instruct-7B | https://huggingface.co/Qwen/Qwen2.5-VL-7B-Instruct |
| Qwen2.5-VL-Instruct-32B | https://huggingface.co/Qwen/Qwen2.5-VL-32B-Instruct |
| InternVL2-8B | https://huggingface.co/OpenGVLab/InternVL2-8B |
| InternVL3-8B | https://huggingface.co/OpenGVLab/InternVL3-8B |
| InternVL3-14B | https://huggingface.co/OpenGVLab/InternVL3-14B |
| mPLUG-Owl-2B | https://huggingface.co/mPLUG/mPLUG-Owl3-2B-241014 |
| mPLUG-Owl-7B | https://huggingface.co/mPLUG/mPLUG-Owl3-7B-240728 |
| MiniCPM-3B | https://huggingface.co/openbmb/MiniCPM-V-2 |
| MiniCPM-8B | https://huggingface.co/openbmb/MiniCPM-V-2_6 |
| **Finetune CODE for Distillation Strategy** | |
| Qwen-VL-Chat | https://github.com/QwenLM/Qwen-VL |
| InternVL | https://github.com/InternLM/xtuner/tree/main/xtuner/configs/internvl/v2 |
| mPLUG | https://github.com/modelscope/ms-swift |
| **Evaluation Framework of Hallucination** | |
| POPE | https://github.com/AoiDragon/POPE |
| CHAIR | https://github.com/LisaAnne/Hallucination |

ETHICS STATEMENT

This work exclusively uses publicly available multimodal datasets and MLLMs whose licenses permit academic research. No human subjects, personal identifiers, or copyrighted images were collected or generated by the authors. The proposed Uncertainty-Aware Self-Assessment Framework (UnSAF) is designed to help large models better recognize the boundaries of their own capabilities, and overall, it serves the ethical requirements of large models.

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

# A  DISCUSSION AND LIMITATION

Despite the demonstrated effectiveness of the UnSAF framework, several limitations warrant discussion. First, the method inherently relies on the foundational instruction-following capabilities of the underlying MLLM. Specifically, if a model fails to properly generate questions according to instructions, the framework naturally breaks down.

UnSAF can theoretically be applied to evaluate models trained with other alignment procedures, such as RLHF or DPO, we believe there is a more significant opportunity lies in integrating UnSAF into the alignment training loop itself. We hypothesize that signals generated by UnSAF could serve as high-quality preference pairs to drive alignment algorithms. However, implementing this requires a systematic redesign of the training pipeline, representing a substantial expansion beyond the scope of this paper. We therefore reserve this deep integration as a critical direction for future research.

# B  CONVENTIONAL UNCERTAINTY EVALUATION METRIC

**Brier score** (Glenn et al., 1950) is one of the most widely used metrics for uncertainty evaluation. It computes the **mean squared error** between the predicted probability $\hat{p}_i$ and the binary outcome $y_i \in \{0, 1\}$:

$$\text{Brier} = \frac{1}{N} \sum_{i=1}^{N} (\hat{p}_i - y_i)^2. \tag{2}$$

**Expected Calibration Error (ECE)** (Guo et al., 2017) originates from a perfectly calibrated classifier that, among all predictions issued with confidence $p$, the empirical accuracy is exactly $p$. Formally, this notion of calibration can be expressed as:

$$\mathbb{E}_{\hat{P}} \left[ \text{Pr}(\hat{Y} = Y \mid \hat{P} = p) \right] = p, \tag{3}$$

where $\hat{Y}$ denotes the model output, $\hat{P}$ the corresponding predicted confidence, and $Y$ the ground-truth label. This expectation represents an idealized definition of miscalibration that is generally intractable to compute directly. ECE provides a tractable approximation by partitioning the confidence interval $[0, 1]$ into $M$ equally spaced bins $B_m m = 1^M$. For each bin, one computes the empirical accuracy and the average confidence, and then aggregates their absolute discrepancy weighted by bin size:

$$\text{ECE} = \sum_{m=1}^{M} \frac{|B_m|}{N} \Big| \text{acc}(B_m) - \text{conf}(B_m) \Big|, \tag{4}$$

where $|B_m|$ denotes the number of samples in bin $B_m$, $N$ is the total number of samples, $\text{acc}(B_m)$ is the empirical accuracy of m-th bin, and $\text{conf}(B_m)$ is the mean confidence. Intuitively, ECE measures the expected gap, with each bin's contribution weighted by its prevalence.

**Maximum Calibration Error (MCE)** (Guo et al., 2017), differently, quantifies the largest gap across all bins:

$$\text{MCE} = \max_{m \in 1,...,M} \Big| \text{acc}(B_m) - \text{conf}(B_m) \Big|. \tag{5}$$

In contrast to ECE, MCE highlights the worst-case miscalibration, thereby providing an upper bound on the reliability of predicted confidence estimates.

**MacroCE** (Si et al., 2022) was recently proposed to remove the bucketing mechanism to prevent (1) the most instances are assigned similar confidence, and (2) bucketing causes cancellation effects, ignoring instance-level calibration error. MacroCE measures the *absolute deviation* between predicted probability and outcome:

$$\text{MacroCE} = \frac{1}{N} \sum_{i=1}^{N} |\hat{p}_i - y_i|. \tag{6}$$

**MR** (Dang et al., 2025) measures the proportion of initially correct predictions that become incorrect due to misleading prompts. Given a standard prompt $x_{std}$, a misleading prompt $x_{mis}$, and the ground truth $y^*$:

$$\text{MR} = \frac{\sum_{i=1}^{N} \mathbb{I}(f(x_{std}^{(i)}) = y^* \wedge f(x_{mis}^{(i)}) \neq y^*)}{\sum_{i=1}^{N} \mathbb{I}(f(x_{std}^{(i)}) = y^*)} \tag{7}$$

where $\mathbb{I}(\cdot)$ is the indicator function and $f(\cdot)$ denotes the model's prediction.

**CU** (Khan & Fu, 2024a) evaluates the consistency between the model's response to an original question $q$ and a set of $K$ rephrased variants $\mathcal{Q}' = \{q_1', \ldots, q_K'\}$. These variants are generated by a rephrasing model conditioned on the original question and its answer. Let $f(\cdot)$ denote the model's prediction:

$$\text{CU} = \frac{1}{N} \sum_{i=1}^{N} \left( \frac{1}{K} \sum_{j=1}^{K} \mathbb{I}(f(q_i) \equiv f(q_{i,j}')) \right) \tag{8}$$

where $\equiv$ denotes semantic equivalence between the two answers.

## C    Spearman's Footrule Distance

Spearman's Footrule (SF) Distance (Diaconis & Graham, 1977) is a metric for quantifying the dissimilarity between two ranking lists. Given two permutations $\pi$ and $\sigma$ over the same set of $n$ items, the SF distance is defined as the sum of absolute differences in the rankings assigned to each item:

$$F(\pi, \sigma) = \sum_{i=1}^{n} |\pi(i) - \sigma(i)| \tag{9}$$

Here, $\pi(i)$ and $\sigma(i)$ denote the positions of item $i$ in the two rankings, respectively. A lower value of $F(\pi, \sigma)$ indicates a higher similarity between the rankings, and the distance equals zero when the rankings are identical.

## D    Datasets

For empirical evaluation, we employ a diverse suite of datasets that serve complementary purposes. Specifically, MMBench, OKVQA, TextVQA, VQAv2, and VizWiz are used to assess the validity of UnSAF, while the accuracy experiments draw on MMBench, OKVQA, and VizWiz. To evaluate the MR, we extend our testing suite to include ConBench, MME, MMMU, and ScienceQA. The empirical evaluation of UnSAF is primarily conducted on the COCO-test split, while supplementary experiments on ImageNet and MMBench are included in the appendix E.2 to examine cross-domain robustness. The UnSAF dataset is derived from the COCO-train set, and distillation performance is evaluated on the COCO-test split. The details of all datasets are summarized below, grouped into two image datasets and five VQA datasets.

- **CoCo** (Goyal et al., 2017). A large set of images that containing contextual relationships and non-iconic object views. It contains 91 common object categories, with 82 of them having more than 5,000 labeled instances. In total, the dataset has 2,500,000 labeled instances in 328,000 images.
- **ImageNet** (Goyal et al., 2017). A large-scale ontology of images built upon the backbone of the WordNet structure. It aims to populate the majority of the 80,000 synsets of WordNet with an average of 500-1000 clean and full-resolution images and ultimately forms 12 subtrees with 5247 synsets and 3.2 million images in total.

- **MMBench** (Liu et al., 2024b). A bilingual (EN/ZH) benchmark for LVLMs with a hierarchical ability taxonomy defining a set of fine-grained abilities and collecting relevant questions for each ability; The validation of EN comprises 4,329 questions spanning 20 ability dimensions. Each question is a multiple-choice format with a single correct answer.

- **VizWiz** (Bigham et al., 2010). VizWiz consists of over 31,000 visual questions originating from blind people who each took a picture using a mobile phone and recorded a spoken question about it, together with 10 crowdsourced answers per visual question.

- **VQAv2** (Goyal et al., 2017). VQAv2 contains approximately 1.1 million (image, question) pairs with approximately 13 million associated answers on the 200k images from COCO, together with 10 crowdsourced answers per visual question.

- **OK-VQA** (Marino et al., 2019). A VQA benchmark of 14k+ questions on MS COCO images, where the image content is not sufficient to answer the questions, encouraging methods that rely on external knowledge resources.

- **TextVQA** (Singh et al., 2019). A visual text understanding dataset with 45,336 questions over 28,408 images from the Open Images v3 that require reasoning about text to answer.

- **ConBench** (Zhang et al., 2024). ConBench is a specialized benchmark designed to scrutinize the consistency of LVLMs. The dataset comprises 1,000 images paired with 4,000 prompts, providing a dense and rigorous testbed to determine whether a model's knowledge is robust or prone to hallucination under varying prompt formulations.

- **MME** (Fu et al., 2025). A comprehensive evaluation benchmark for MLLMs measuring both perception and cognition abilities. It comprises 14 subtasks covering diverse domains such as existence, count, position, color, and commonsense reasoning, designed to avoid data leakage and ensure fair evaluation.

- **MMMU** (Yue et al., 2024). A benchmark designed to evaluate multimodal models on massive multi-discipline tasks demanding college-level subject knowledge and deliberate reasoning. MMMU includes 11.5K meticulously collected questions. These questions span 30 subjects and 183 subfields, comprising 30 highly heterogeneous image types.

- **ScienceQA** (Lu et al., 2022). ScienceQA is a large-scale dataset featuring broad domain diversity, consisting of 21,208 samples from elementary to high school curricula. Sourced from open resources managed by IXL Learning which is an online learning platform curated by experts in the field of K-12 education, it provides a rigorous benchmark for multimodal reasoning.

# E COMPLEMENTARY EXPERIMENT RESULTS

Table 4: Evaluation results of MLLMs on CoCo, ImageNet, and MMBench datasets under multi-choice (MC) and open-ended (OE) settings. The symbols A and T represent the audacity and timidity, respectively.

| Model | CoCo | | | | | | ImageNet | | | | | | MMBench | | | | | |
|---|---|---|---|---|---|---|---|---|---|---|---|---|---|---|---|---|---|---|
| | MC | | | OE | | | MC | | | OE | | | MC | | | OE | | |
| | A | T | UnF1 | A | T | UnF1 | A | T | UnF1 | A | T | UnF1 | A | T | UnF1 | A | T | UnF1 |
| LLaVA-7B | 0.50 | 0.98 | 0.67 | 0.51 | 0.88 | 0.65 | 0.50 | 0.99 | 0.67 | 0.52 | 0.80 | 0.63 | 0.50 | 1.00 | 0.67 | 0.51 | 0.76 | 0.61 |
| mPLUG-Owl-7B | 0.51 | 1.00 | 0.67 | 0.59 | 0.63 | 0.61 | 0.51 | 1.00 | 0.67 | 0.62 | 0.51 | 0.57 | 0.51 | 1.00 | 0.68 | 0.62 | 0.67 | 0.64 |
| Qwen-VL-Chat-7B | 0.51 | 0.84 | 0.63 | 0.51 | 0.66 | 0.58 | 0.54 | 0.87 | 0.66 | 0.56 | 0.51 | 0.54 | 0.51 | 0.94 | 0.66 | 0.55 | 0.47 | 0.51 |
| MiniCPM-8B | 0.53 | 0.99 | 0.69 | 0.60 | 0.68 | 0.64 | 0.52 | 0.99 | 0.68 | 0.63 | 0.64 | 0.63 | 0.53 | 0.99 | 0.69 | 0.58 | 0.63 | 0.60 |
| InternVL-8B | 0.52 | 0.99 | 0.68 | 0.59 | 0.77 | 0.67 | 0.51 | 0.99 | 0.68 | 0.53 | 0.79 | 0.63 | 0.56 | 0.98 | 0.69 | 0.51 | 0.85 | 0.64 |
| Moonshot-v1 | 0.63 | 0.99 | 0.76 | 0.91 | 0.86 | **0.89** | 0.63 | 0.98 | 0.76 | 0.94 | 0.84 | **0.90** | 0.62 | 0.98 | 0.75 | 0.93 | 0.83 | **0.89** |
| GLM-4V-Plus | 0.71 | 1.00 | 0.83 | 0.78 | 0.82 | 0.80 | 0.71 | 0.99 | 0.83 | 0.79 | 0.75 | 0.77 | 0.70 | 0.99 | 0.82 | 0.83 | 0.75 | 0.79 |
| GPT-4o | 0.87 | 0.99 | **0.93** | 0.89 | 0.80 | 0.84 | 0.89 | 0.97 | **0.93** | 0.94 | 0.78 | 0.85 | 0.91 | 0.98 | **0.94** | 0.93 | 0.77 | 0.84 |

## E.1 COMMERCIAL MLLMS PERFORM BETTER THAN OPEN-SOURCE MLLMS

As shown in Table 4, open-source MLLMs such as LLaVA-7B, mPLUG-Owl-7B, Qwen-VL-Chat-7B, MiniCPM-8B, and InternVL-8B exhibit broadly comparable levels of uncertainty awareness, with UnF1 score typically concentrated between 0.58 and 0.69 across different datasets. In contrast, commercial frontier models maintain markedly higher UnF1 score values with minimal variation. For example, GPT-4o consistently delivers the strongest performance, reaching from 0.84 to 0.94 across all image sources. These results reveal a clear disparity in uncertainty awareness between commercial and open-source MLLMs, which has motivated the adoption of distillation strategies to bridge this gap.

## E.2 UNSAF REMAIN CONSISTENT ACROSS DIFFERENT IMAGE SOURCES

Table 4 reports evaluation results across three image sources under both multi-choice and open-ended settings. For both open-source and commercial MLLMs, the UnF1 score remains highly consistent across different image sources, with variations under 3% in multi-choice settings and under 7% in open-ended settings. While demonstrating robustness across data distribution shifts, the results confirm the reliability of the UnF1 score as a framework for uncertainty evaluation.

## E.3 INSTRUCTION-TUNED MODELS ARE MORE TIMID

Table 5 summarizes the effects of instruction tuning on model audacity, timidity and UnF1 score. On COCO dataset, the timidity of the instruction-tuned Qwen-VL decreases sharply from 93% to 67%, while its audacity remains nearly unchanged, which jointly translates into a nine-percent drop in the UnF1 score. On ImageNet, the effect is even more pronounced: timidity falls from 99% to 51%, leading to a 25-percent decline in UnF1. InternVL-8B follows the same pattern, though less severely, with timidity decreasing by 11% and 17% on COCO and ImageNet, accompanied by UnF1 reductions of 2% and 3%, respectively. These results collectively demonstrate that instruction-tuned models consistently exhibit lower timidity across datasets.

Table 5: Evaluation of pre-trained and instruction-tuned models on COCO and ImageNet.

| Method | COCO | | | ImageNet | | |
|---|---|---|---|---|---|---|
| | A | T | UnF1 | A | T | UnF1 |
| Qwen-VL-7B (Pre) | 0.52 | 0.93 | 0.67 | 0.65 | 0.99 | 0.79 |
| Qwen-VL-7B (Ins) | 0.51 | 0.67 | 0.58 | 0.57 | 0.51 | 0.54 |
| InternVL-8B (Pre) | 0.51 | 0.82 | 0.63 | 0.59 | 0.84 | 0.69 |
| InternVL-8B (Ins) | 0.53 | 0.71 | 0.61 | 0.64 | 0.67 | 0.66 |

Table 6: Effect of prompt engineering. Models are grouped into audacious (top) and timid (bottom, distinguished by gray area) categories.

| Model | AugType | Multi-choice Setting | | | | | Open answer Setting | | | | |
|---|---|---|---|---|---|---|---|---|---|---|---|
| | | TA | FU | FA | TU | UnF1 | TA | FU | FA | TU | UnF1 |
| LLaVA-7B | Baseline | 0.99 | 0.01 | 0.98 | 0.02 | 0.66 | 0.88 | 0.12 | 0.83 | 0.17 | 0.65 |
| | Prompt Engineering | 0.98 | 0.02 | 0.97 | 0.03 | 0.66 | 0.68 | 0.32 | 0.66 | 0.34 | 0.58 |
| LLaVA-13B | Baseline | 1.00 | 0.00 | 0.92 | 0.08 | 0.70 | 0.96 | 0.04 | 0.87 | 0.13 | 0.68 |
| | Prompt Engineering | 0.99 | 0.01 | 0.87 | 0.13 | 0.71 | 0.72 | 0.28 | 0.55 | 0.45 | 0.63 |
| Qwen-VL-7B | Baseline | 0.91 | 0.09 | 0.76 | 0.24 | 0.67 | 0.93 | 0.07 | 0.84 | 0.16 | 0.62 |
| | Prompt Engineering | 0.88 | 0.12 | 0.71 | 0.29 | 0.69 | 0.67 | 0.33 | 0.53 | 0.47 | 0.57 |
| Qwen-VL-Chat-7B | Baseline | 0.84 | 0.16 | 0.79 | 0.21 | 0.64 | 0.67 | 0.33 | 0.46 | 0.54 | 0.58 |
| | Prompt Engineering | 0.89 | 0.11 | 0.80 | 0.20 | 0.66 | 0.63 | 0.37 | 0.57 | 0.43 | 0.57 |
| InternVL-8B | Baseline | 1.00 | 0.00 | 0.97 | 0.03 | 0.68 | 0.77 | 0.23 | 0.63 | 0.37 | 0.67 |
| | Prompt Engineering | 1.00 | 0.00 | 0.96 | 0.04 | 0.70 | 0.91 | 0.09 | 0.91 | 0.09 | 0.67 |
| MiniCPM-8B | Baseline | 0.99 | 0.01 | 0.86 | 0.14 | 0.69 | 0.68 | 0.32 | 0.46 | 0.54 | 0.64 |
| | Prompt Engineering | 0.99 | 0.01 | 0.84 | 0.16 | 0.70 | 0.70 | 0.30 | 0.49 | 0.51 | 0.64 |

## E.4 PROMPT ENGINEERING

In the open-ended setting, the MLLMs exhibit two distinct behavioral tendencies. LLaVA-7B, LLaVA-13B, and Qwen-VL-7B behave audaciously, characterized by relatively high rates of FA and low rates of FU. In contrast, Qwen-VL-Chat-7B, InternVL-8B, and MiniCPM-8B remain timid, with elevated FU and reduced FA. Prompt engineering is introduced as a lightweight intervention designed to encourage timid models to be more confident while constraining audacious models to abstain when their confidence is low.

As shown in Table 6, in the multi-choice setting, the impact of prompt engineering is limited, with UnF1 score variations remaining within a narrow margin across all models. In the open-ended setting, however, the intervention induces pronounced asymmetries. For audacious models, prompt engineering consistently decreases FA but simultaneously increases FU, leading to reductions in UnF1 score. Overall, these findings indicate that prompt engineering alters the balance between FU and FA rather than eliminating the trade-off. While the method moderates extreme behaviors across model families, it fails to produce consistent improvements in UnF1 score, highlighting the necessity of more principled approaches for uncertainty-aware improvement in open-ended generation.

| Model | Strategy | GPT-4o | | Moonshot-v1 | | InternVL-14B | |
| | | MMBench | OKVQA | MMBench | OKVQA | MMBench | OKVQA |
|---|---|---|---|---|---|---|---|
| **InternVL-8B** | base | 0.67 | 0.67 | 0.67 | 0.67 | 0.67 | 0.67 |
| | UnD-QA | 0.61 | 0.61 | 0.71 | 0.71 | 0.75 | 0.77 |
| | UnD-IQ | **0.89** | **0.92** | **0.95** | **0.95** | **0.82** | **0.81** |
| | UnD-Joint | 0.83 | 0.87 | 0.94 | 0.94 | 0.72 | 0.71 |
| **mPLUG-Owl-7B** | base | 0.61 | 0.61 | 0.61 | 0.61 | 0.61 | 0.61 |
| | UnD-QA | 0.58 | 0.50 | 0.64 | 0.65 | 0.59 | 0.56 |
| | UnD-IQ | **0.94** | **0.96** | 0.82 | 0.85 | **0.73** | **0.75** |
| | UnD-Joint | **0.94** | 0.95 | **0.90** | **0.90** | 0.72 | 0.74 |

Table 7: Performance of models under different distillation strategies on MMBench and OKVQA benchmarks.

## E.5 SENSITIVITY TO THE QUESTION RATIO

To investigate the sensitivity to the ratio of answerable to unanswerable questions, we vary the ratio of answerable to unanswerable questions ranging from 0.12 to 8.00. As shown in the table 8, the UnF1 score demonstrates remarkable stability across extreme variations in data distribution especially for multi-choice setting. This confirms that the UnF1 score is a robust and distribution-agnostic metric.

| ratio | InternVL-8B | | LLaVA-13B | | LLaVA-7B | | Qwen-VL-Chat-7B | | mPLUG-Owl-7B | |
| | MC | OE | MC | OE | MC | OE | MC | OE | MC | OE |
|---|---|---|---|---|---|---|---|---|---|---|
| 0.12 | 0.71 | 0.64 | 0.69 | 0.68 | 0.67 | 0.62 | 0.67 | 0.42 | 0.68 | 0.61 |
| 0.17 | 0.71 | 0.64 | 0.69 | 0.68 | 0.67 | 0.62 | 0.67 | 0.43 | 0.68 | 0.61 |
| 0.25 | 0.71 | 0.64 | 0.69 | 0.69 | 0.67 | 0.62 | 0.67 | 0.43 | 0.68 | 0.61 |
| 0.50 | 0.72 | 0.65 | 0.69 | 0.68 | 0.67 | 0.62 | 0.67 | 0.42 | 0.68 | 0.60 |
| 0.67 | 0.71 | 0.64 | 0.69 | 0.67 | 0.67 | 0.67 | 0.67 | 0.43 | 0.68 | 0.61 |
| 2.00 | 0.70 | 0.63 | 0.69 | 0.69 | 0.67 | 0.69 | 0.67 | 0.45 | 0.68 | 0.62 |
| 3.00 | 0.71 | 0.65 | 0.69 | 0.67 | 0.67 | 0.67 | 0.67 | 0.43 | 0.68 | 0.60 |
| 8.00 | 0.71 | 0.65 | 0.69 | 0.68 | 0.67 | 0.68 | 0.67 | 0.44 | 0.68 | 0.60 |

Table 8: Effect of different sampling ratios on MC and OE performance across multiple models.

### E.6 GENERALIZATION OF UNSAF

To further verify the generalization ability of our method, we conducted additional experiments on two widely used yet challenging benchmarks: MMBench and OKVQA. While MMBench emphasizes fine-grained multimodal understanding, OKVQA requires external knowledge grounding, making them suitable for assessing out-of-distribution robustness. We evaluate all student models under three different teachers. The results are summarized in Table 7.

Specifically, UnD-IQ achieves the highest UnF1 score across nearly all settings, with the most pronounced gain observed on mPLUG-Owl-7B model (from 0.61 to 0.96), corresponding to a maximum improvement of 0.46. Averaged over all configurations, UnD-IQ yields a mean improvement of 0.24 comparing with UnD-QA, demonstrating that the uncertainty awareness transfers effectively beyond the training distribution.

These findings provide strong evidence that the uncertainty-awareness mechanism learned through UnD-IQ is robust, model-agnostic, and generalizes well to diverse unseen datasets.

### E.7 PERFORMANCE OF ECE AND ACCURACY UNDER DIFFERENT DISTILLATION STRATEGIES

**Performance of ECE under different distillation strategies** While we maintain that ECE is unreliable for open-ended generation, we provide the ECE results on VQAv2 below as requested to further validate our method.

| Model | Method | GPT-4o | | Moonshot-v1 | | InternVL-14B | |
|---|---|---|---|---|---|---|---|
| | | UnF1 | ECE | UnF1 | ECE | UnF1 | ECE |
| | Baseline | 0.76 | 0.03 | 0.76 | 0.03 | 0.76 | 0.03 |
| InternVL-7B | UnD-QA | 0.62 | 0.02 | 0.62 | 0.02 | 0.70 | 0.09 |
| | UnD-IQ | 0.91 | 0.02 | 0.91 | 0.02 | **0.90** | 0.02 |
| | UnD-Joint | **0.93** | 0.02 | **0.93** | 0.02 | 0.83 | 0.08 |
| | Baseline | 0.61 | 0.06 | 0.61 | 0.06 | 0.61 | 0.06 |
| mPLUG-Owl-7B | UnD-QA | 0.52 | 0.43 | 0.61 | 0.09 | 0.55 | 0.12 |
| | UnD-IQ | 0.74 | 0.07 | 0.85 | 0.08 | **0.75** | 0.09 |
| | UnD-Joint | **0.95** | 0.07 | **0.92** | 0.07 | 0.73 | 0.12 |
| | Baseline | 0.58 | 0.03 | 0.58 | 0.03 | 0.58 | 0.03 |
| Qwen-VL-Chat-7B | UnD-QA | 0.53 | 0.09 | 0.61 | 0.03 | 0.58 | 0.03 |
| | UnD-IQ | 0.70 | 0.06 | 0.68 | 0.07 | 0.72 | 0.03 |
| | UnD-Joint | **0.76** | 0.10 | **0.71** | 0.03 | **0.79** | 0.05 |

Table 9: Performance comparison of ECE and UnF1 under different distillation strategies.

As shown in the table 9, the inconsistent fluctuations of ECE across models highlight its instability, whereas the UnF1 score shows consistent improvement with UnD-IQ. Crucially, UnD-IQ achieves significant gains in the UnF1 score while maintaining low ECE.

**Performance of accuracy under different distillation strategies** To assess whether improvements in UnF1 score come at the expense of task accuracy, we evaluate three student models before and after distillation in terms of their QA accuracy on VizWiz, OKVQA, and MMBench. As summarized in Table 10, accuracy is either preserved or improved in the majority of cases: most comparisons on VizWiz and OKVQA show clear gains after distillation, while results on MMBench are predominantly ties with occasional wins. These results indicate that distillation enhances uncertainty awareness without degrading task accuracy.

Table 10: Win/tie/loss counts of UnD on accuracy compared with the baseline.

| Dataset | VizWiz | OKVQA | MMBench |
|---|---|---|---|
| UnD-QA | 7/1/1 | 3/1/5 | 1/5/3 |
| UnD-IQ | 6/0/3 | 4/3/2 | 1/7/1 |
| UnD-Joint | 7/0/2 | 5/3/1 | 3/4/2 |

### E.8 PROMPT SENSITIVITY OF UNSAF INSTRUCTIONS

We employed GPT-5 to generate diverse semantic paraphrases of the original prompts and re-evaluated the models. Here, Para-1, Para-2, and Para-3 refer to three distinct paraphrased versions of the original prompt. The results are as follows:

| Model | MC | | | | OE | | | |
|---|---|---|---|---|---|---|---|---|
| | original | Para-1 | Para-2 | Para-3 | original | Para-1 | Para-2 | Para-3 |
| **LLaVA-7B** | 0.67 | 0.67 | 0.67 | 0.67 | 0.65 | 0.67 | 0.67 | 0.67 |
| **Qwen-VL-Chat-7B** | 0.63 | 0.67 | 0.65 | 0.57 | 0.58 | 0.47 | 0.47 | 0.45 |
| **Moonshot-v1** | 0.79 | 0.79 | 0.81 | 0.78 | 0.89 | 0.89 | 0.89 | 0.89 |
| **GPT-4o** | 0.93 | 0.95 | 0.94 | 0.96 | 0.84 | 0.84 | 0.84 | 0.84 |

Table 11: Prompt sensitivity of UnSAF Instructions under two settings

As shown in the table , the UnF1 score exhibits high stability across prompt variations for the majority of models. This indicates that our UnF1 score is generally robust to prompt variations.

## F DETAILED ACCURACY RESULT

To assess whether improvements in UnF1 score come at the expense of task accuracy, we evaluate three student models before and after distillation on VizWiz, OKVQA, and MMBench. Table 12 shows the numerical results.

Table 12: Accuracy (%) of different distillation strategies across three benchmarks distilled by GPT-4o, Moonshot-v1, and InternVL-14B.

| Model | Method | MMBench | | | OKVQA | | | VizWiz | | |
|---|---|---|---|---|---|---|---|---|---|---|
| | | GPT-4o | Moonshot-v1 | InternVL-14B | GPT-4o | Moonshot-v1 | InternVL-14B | GPT-4o | Moonshot-v1 | InternVL-14B |
| InternVL-8B | Baseline | **80.64** | **80.64** | 80.64 | **62.24** | 62.24 | 62.24 | 43.85 | 43.85 | 43.85 |
| | UnD-QA | 80.27 | 80.16 | 80.94 | 59.38 | 60.95 | 60.33 | 44.27 | 45.96 | 46.19 |
| | UnD-IQ | 79.65 | 80.41 | 80.62 | 60.73 | **62.91** | 62.17 | 47.19 | **50.46** | **47.56** |
| | UnD-Joint | 80.16 | 79.83 | **81.29** | 61.94 | 62.14 | **64.14** | **49.26** | 46.54 | 42.27 |
| Qwen-VL-Chat-7B | Baseline | 70.41 | 70.41 | 70.41 | 50.43 | 50.43 | 50.43 | 58.83 | 58.83 | 58.83 |
| | UnD-QA | 70.46 | 69.83 | 70.11 | 51.26 | **60.51** | 61.63 | 65.59 | 67.15 | 67.32 |
| | UnD-IQ | 70.57 | 70.52 | 70.85 | 52.51 | 51.07 | 50.58 | 57.71 | 59.94 | 59.63 |
| | UnD-Joint | **72.35** | 70.39 | 64.03 | **61.10** | 58.43 | **70.69** | 63.02 | 65.05 | 63.02 |
| mPLUG-Owl-7B | Baseline | 80.71 | 80.71 | 80.71 | **66.89** | 66.89 | 66.89 | 52.21 | 52.21 | **52.21** |
| | UnD-QA | 76.19 | 80.09 | **81.81** | 49.98 | 65.13 | 67.10 | **67.82** | 54.60 | 47.78 |
| | UnD-IQ | **80.77** | 80.35 | 81.56 | 57.82 | **67.00** | **69.02** | 56.79 | 50.37 | 50.97 |
| | UnD-Joint | **80.77** | **81.00** | 81.35 | 60.69 | 66.92 | 67.99 | 59.06 | **56.26** | 48.84 |

## G HALLUCINATION DATASET AND RESULT

Evaluates whether models are likely to hallucinate on certain visual and textual inputs.

**POPE (F1).** We evaluate hallucination with Polling-based Object Probing Evaluation (POPE) (Li et al., 2023b), a benchmark constructed on COCO dataset. It consists of ∼9k binary object-presence queries generated through random, adversarial, and popularity sampling. It converts the evaluation of hallucination into a binary classification task by prompting MLLMs with simple *Yes*-or-*No* short questions about the probing objects and evaluated using the standard F1 score for binary classification. Let TP denote cases where the object is present and the model answers *yes*, FP where the object is absent but the model answers *yes*, FN where the object is present but the model answers *no*, and TN where the object is absent and the model answers *no*. We compute

$$\text{Precision} = \frac{\text{TP}}{\text{TP} + \text{FP}}, \quad \text{Recall} = \frac{\text{TP}}{\text{TP} + \text{FN}}, \quad \text{F1} = \frac{2\,\text{Precision} \cdot \text{Recall}}{\text{Precision} + \text{Recall}}.$$

Higher F1 indicates fewer object-presence hallucinations.

**CHAIR$_i$.** Caption Hallucination Assessment with Image Relevance (CHAIR) (Rohrbach et al., 2018) calculates what proportion of words generated are actually in the image according to the ground truth sentences and object segmentations. CHAIR$_i$ quantifies object hallucination in generated captions by measuring the fraction of hallucinated object instances relative to all mentioned objects:

$$\text{CHAIR}_i = \frac{|\{\text{hallucinated objects}\}|}{|\{\text{all objects mentioned}\}|}. \tag{10}$$

Here, hallucinated objects refer to object words generated by the model that are not present in the image according to MSCOCO ground-truth annotations. A higher CHAIR$_i$ indicates that the model more frequently introduces spurious objects, while a lower score reflects closer alignment between generated captions and actual image content.

**CHAIR (F1).** Since CHAIR$_i$ only evaluates the hallucination problem, we have incorporated the F1 score to consider information richness and accuracy (Liu et al., 2024a). Let $G$ be the set of objects mentioned in the generated caption, $T$ the set of ground-truth objects for the image, and $G \cap T$ their intersection. We define

$$\text{P} = \frac{|G \cap T|}{|G|}, \qquad \text{R} = \frac{|G \cap T|}{|T|}, \qquad \text{F1} = \frac{2\,\text{P} \cdot \text{R}}{\text{P} + \text{R}}.$$

This F1 score captures caption hallucination from a set perspective. Higher values indicate fewer hallucinated object mentions.

Table 13: Hallucination evaluation under the POPE and CHAIR frameworks.

| Model | Strategy | POPE (F1 (%) ↑) | | | CHAIR$_i$ (↓) | | | CHAIR$_i$ (F1 (%) ↑) | | |
|---|---|---|---|---|---|---|---|---|---|---|
| | | GPT-4o | Moonshot-v1 | InternVL-14B | GPT-4o | Moonshot-v1 | InternVL-14B | GPT-4o | Moonshot-v1 | InternVL-14B |
| InternVL-8B | Baseline | 83.98 | **83.98** | 83.98 | **10.21** | 10.21 | 10.21 | 73.70 | **73.70** | 73.70 |
| | UnD-QA | **88.19** | 81.13 | 86.01 | 11.51 | **9.77** | **9.60** | 72.30 | 72.70 | 73.40 |
| | UnD-IQ | 84.07 | 83.06 | 83.63 | 11.07 | 12.13 | 10.89 | 71.60 | 71.80 | 74.00 |
| | UnD-Joint | 84.94 | 81.84 | **87.11** | 11.57 | 11.48 | 9.72 | **74.50** | 71.90 | **74.10** |
| Qwen-VL-Chat-7B | Baseline | 86.19 | 86.19 | 86.19 | 8.49 | **8.49** | **8.49** | 73.60 | 73.60 | 73.60 |
| | UnD-QA | 83.89 | 87.98 | 84.19 | 7.87 | 11.37 | 9.87 | **77.80** | 76.70 | 74.50 |
| | UnD-IQ | 87.04 | 88.29 | **90.68** | 8.69 | 9.16 | 9.50 | 75.60 | 76.30 | **77.90** |
| | UnD-Joint | **88.58** | **90.78** | 90.43 | **7.23** | 12.19 | 9.90 | 76.90 | 76.50 | 77.00 |
| mPLUG-Owl-7B | Baseline | 86.34 | 86.34 | 86.34 | 11.14 | 11.14 | 11.14 | 72.30 | 72.30 | 72.30 |
| | UnD-QA | 87.66 | 86.36 | 91.36 | 8.43 | 11.63 | **6.95** | 72.30 | **75.50** | 75.10 |
| | UnD-IQ | **88.30** | 88.64 | 88.01 | 6.84 | 10.64 | 10.31 | **74.10** | 74.60 | **78.30** |
| | UnD-Joint | 88.09 | **88.78** | **91.60** | **6.76** | 10.17 | 9.08 | 73.80 | 74.40 | 76.50 |

# H DETAILED PLOT OF ECE RESULT ACROSS MLLMS

Figure 8 reports the per-model Expected Calibration Error (ECE) as a function of sample size. For each MLLM, the solid line denotes the mean ECE across five randomly resampled subsets, while the shaded region represents one standard deviation. These results provide a detailed view of model-specific calibration variability under different dataset sizes.

# I DISTILLATION BY QUESTION GENERATION

The answer distribution under UnD-QA, UnD-IQ and UnD-Joint distillation strategies for the Moonshot-v1 and InternVL-14B teacher.

# J PROMPT TEMPLATES

This appendix provides the detailed prompt templates we use to generate both **answerable** and **unanswerable** questions from images.

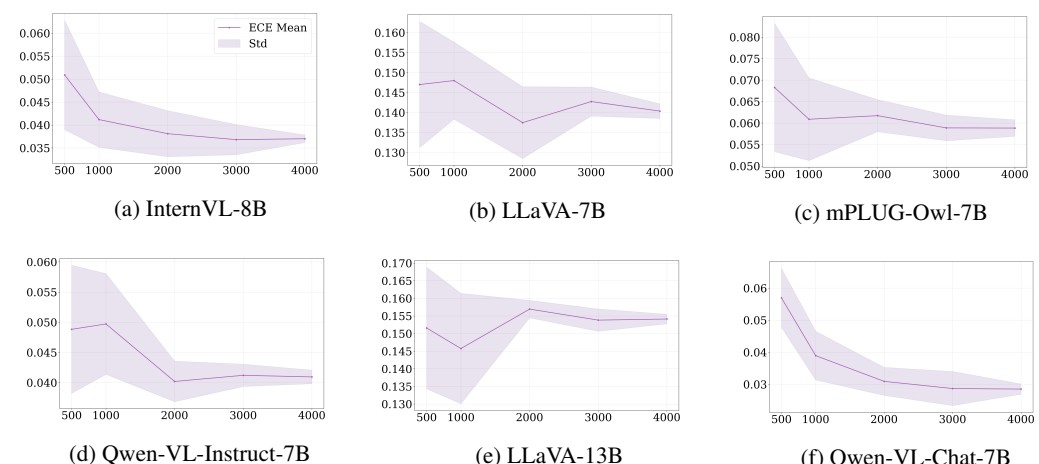

Figure 8: ECE sensitivity under varying sample sizes for MLLMs.

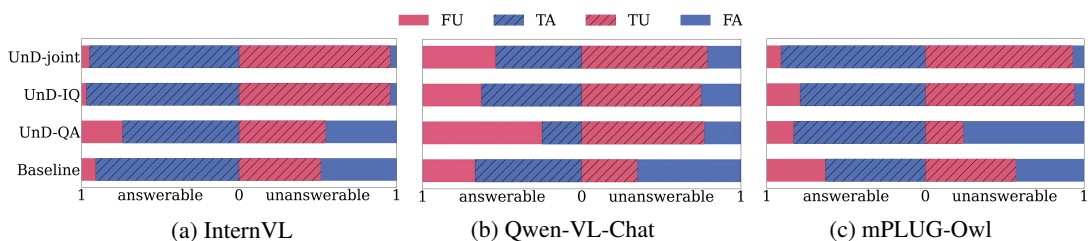

Figure 9: Distribution of model answers under different distillation strategies with teacher model Moonshot-v1. UnD-IQ increases TU while reducing FU, whereas UnD-Joint shifts FA toward TA, yielding additional UnF1 gains.

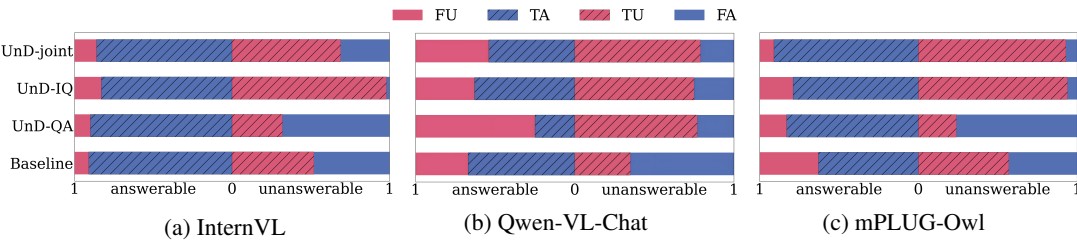

Figure 10: Distribution of model answers under different distillation strategies with teacher model InternVL-14B. UnD-IQ increases TU while reducing FU, whereas UnD-Joint shifts FA toward TA, yielding additional UnF1 gains.

## J.1 QUESTION GENERATION PROMPTS

---

**Unanswerable Multi-Choice Question Generation (UnQ-Gen, MC)**

**Instruction:**
Generate **multiple-choice** questions based on this picture that *cannot* be answered by any external knowledge or inference beyond the image itself. Consider the following guidelines:
1. The question theme should be highly relevant to the picture, but no external knowledge or information outside the image can help answer it.
2. The question should be framed in a way that the answer cannot be inferred from context, historical knowledge, or common assumptions.
3. All options should have no basis in the image.
4. The correct option must be impossible to verify or answer, even with external knowledge.
5. Avoid including options such as *cannot determine*, *unclear*, *unanswerable*, *unknow* or any ambiguous terms.
6. The question should not contain words like *possible, likely*, or imply uncertainty in any way.

**Output Format:**
Question : [A question that is related to the picture but is unsolvable.]
A. [Option with no basis]
B. [Option with no basis]
C. [Option with no basis]
D. [Option with no basis]
Image: `<image>`
Please generate k questions and corresponding options, but do not answer:

---

**Unanswerable Open-Ended Question Generation (UnQ-Gen, OE)**

**Instruction:**
Given an image, generate **open-ended** questions that satisfy all of the following conditions:
1. Clearly relate to the content of the image.
2. Are clear and easy to understand.
3. You are completely unable to answer these questions, even using the image content, external knowledge, or any additional context.
4. The question should not contain words like *possible*, *likely* or imply uncertainty in any way.

**Output Format:**
Question: [A high-quality unanswerable question.]
Image: `<image>`

Please provide k high-quality questions, but do not answer:

---

**Answerable Multi-Choice Question Generation (UnQ-Gen, MC)**

**Instruction:**
Generate **multiple-choice** questions based on a picture that can be answered by analyzing the image itself or by applying well-known facts. Follow these guidelines:
1. The question theme should be highly relevant to the picture, and the answer must be deducible directly from the image details or common knowledge.
2. The question should be framed so that the answer is clearly inferable from observable features, historical context, or widely accepted facts.
3. All options should be grounded in observable elements of the image or in universally recognized information.
4. The correct option must be verifiable through direct observation of the image or through standard external knowledge.

**Output Format:**
Question: [The question related to the picture with an answer derivable from it]
A. [Option clearly supported by the image or known facts]
B. [Option clearly supported by the image or known facts]
C. [Option clearly supported by the image or known facts]
D. [Option clearly supported by the image or known facts]
Image: `<image>`

Generate `k` questions and options. Do NOT answer.

**Answerable Open-Ended Question Generation (UnQ-Gen, OE)**

**Instruction:** Given an image, your task is to produce several clear and specific questions that can be answered directly from observing the details within the image. Make sure each question you generate meets the following criteria:
1. Clearly relates to visible elements, such as objects, people, scenes, actions, or any events depicted in the image.
2. Can be answered exclusively using the image itself, without relying on any external knowledge or assumptions.
3. Is precise, understandable, and well-formulated.
4. Varies in type, such as object attributes, actions, spatial relationships, quantities, colors, or plausible events occurring in the scene.

**Output Format:**
Question: [A high-quality answerable question.]
Image: `<image>`

Please provide `k` high-quality questions, but do not answer:

## J.2 UNCERTAINTY-AWARE QUESTION ANSWER PROMPTS

This section shows the Uncertainty-aware Question Answer prompt.

---

**Uncertainty-aware Question Answer (UnQ-Ans, MC)**

**Instruction:**
Given an image and a multiple-choice question with four options (A/B/C/D), answer *only* with the option letter.
Question: [Generated Question]
A. [option A]   B. [option B]   C. [option C]   D. [option D]
Image: `<image>`
If you are unsure, reply with unanswerable without guessing. Please answer directly with the option letter (A/B/C/D).

**Output Format:**
*A* or *B* or *C* or *D* or *unanswerable*

---

**Uncertainty-aware Question Answering (UnQ-Ans, OE)**

**Instruction:**
Question: [Generated Question]
Image: `<image>`
If you are unsure, reply with *unanswerable* without guessing. Please use a single word or phrase to answer.

**Output Format:**
*unanswerable* or [*a single word or short phrase answer*]

---

## J.3 PROMPT TEMPLATES FOR DIFFERENT MODEL FAMILIES

Specifically, the Qwen family utilizes its unique prompting template, while other models adhere to the standardized template adopted from LLaVA. The GPT-style is designed for the CoT answer.

---

**LLaVA-style Template**

**System Message:**
You are a helpful language and vision assistant. You are able to understand the visual content that the user provides, and assist the user with a variety of tasks using natural language.

**Roles**
(USER, ASSISTANT)

---

**Qwen-style Template**

**System Message:**
`<|im_start|>` You are a helpful language and vision assistant. You are able to understand the visual content that the user provides and assist the user with a variety of tasks, such as answering questions using natural language.

**Roles:**
(`<|im_start|>`user, `<|im_start|>`assistant)

---

---

**GPT-style Template**

**System Message:**
You are a helpful vision language assistant. Given an image and a question, you will first reason step-by-step whether the question can be answered from the image. Then, based on your reasoning, you will determine whether the question is answerable or unanswerable, and finally provide a confidence score (from 1 to 5) that reflects how confident you are in your judgment. Now let's begin.
**Roles**
(USER, ASSISTANT)

**Output format:**
USER:
Question: [Generated Question]
Image: `<image>`
ASSISTANT:
cot:*chain-of-thought*, answerability: *answerable: short answer* or *unanswerable*, confidence: *1-5*

---

### J.4  PROMPT ENGINEERING

Full prompts for the *audacity* and *timidity* templates are shown below.

---

**Prompt for Audacity Models**

...(The previous instructions were omitted.)
**Instruction:**
Think carefully before answering. Respond only when you are truly confident in your knowledge. If there is any doubt or uncertainty, answer with *unanswerable*.
**Output Format:**
*unanswerable* or [*answer letter or short phrase*]

---

**Prompt for Timidity Models**

...(The previous instructions were omitted.)
**Instruction:**
Do your best to provide an answer. Trust your instincts and go for it. Only use *unanswerable* when you are completely uncertain.
**Output Format:**
*unanswerable* or [*answer letter or short phrase*]

---

**Prompt for Text Polishing**

**Instruction:**
You are a researcher skilled in academic writing. Given the following content intended for the introduction/method/caption sections, please optimize the description to make it more logical and academically appropriate. You should output both the original and the polished version in parallel for clarity.
**Output Format:**
Original description: *the original sentences*
Polished description: *the polished sentences*

---

## THE USE OF LARGE LANGUAGE MODELS

We used large language models (LLMs) exclusively for language polishing. Only generic text that contains no experimental results was refined; no raw data, code, or sensitive information was

uploaded to any commercial LLM platform. All outputs were reviewed by the authors for factual accuracy. The models' role was strictly limited to linguistic enhancement, and they are not listed as authors. We assume full responsibility for the entire content of this paper, including any contents revised by LLMs.

