# OpenReview forum: "UnSAF: A Self-Assessment Framework of Uncertainty Awareness for Multimodal LLMs"
_ICLR.cc/2026/Conference — Submitted to ICLR 2026_

### Official Review · Reviewer_3F9m · 2025-10-26

**Soundness:** 2
**Presentation:** 3
**Contribution:** 2
**Rating:** 4
**Confidence:** 4

**Summary:**

This paper addresses the critical limitation of MLLMs, their tendency to generate plausible yet incorrect responses (hallucinations), by proposing the Uncertainty-Aware Self-Assessment Framework (UnSAF). Unlike conventional uncertainty evaluation metrics that rely on extensive labeled datasets and struggle with open-ended multimodal tasks, UnSAF operates in two stages: first, prompting MLLMs to generate both answerable and unanswerable questions grounded in given images, then requiring models to answer these self-generated questions. Responses are categorized into four types (True Answerable, False Answerable, True Unanswerable, False Unanswerable) to compute an interpretable, label-free Uncertainty-aware F1 (UnF1) score. The paper further explores a knowledge distillation framework (with three variants: UnD-QA, UnD-IQ, UnD-Joint) to enhance uncertainty awareness in small-scale open-source MLLMs, showing that distilling uncertainty-aware question generation (UnD-IQ) yields significant UnF1 improvements without degrading task performance or increasing hallucinations.

**Strengths:**

1. Label-free uncertainty evaluation paradigm. UnSAF addresses a critical gap in MLLM assessment by eliminating reliance on labeled datasets – a major limitation of metrics like Brier score and ECE. By framing uncertainty evaluation as a self-assessment task (generate questions → answer them → categorize responses), it naturally adapts to open-ended multimodal tasks where exact string matching or predefined labels fail (e.g., "a man riding a bike" vs. "a person on a bicycle").
2. Robustness and consistency across scenarios. Extensive validity analysis shows UnSAF’s UnF1 score maintains stable model rankings across diverse datasets (MMBench, VQAv2, VizWiz, etc.) and is resilient to sample size variations – a stark contrast to conventional metrics (ECE, MCE) that exhibit high volatility, especially with small datasets. This makes UnSAF practical for low-data regimes common in multimodal research.
3. Insightful empirical observations on MLLM behavior. The paper uncovers actionable patterns: (1) instruction-tuned MLLMs exhibit "timid" behavior (over-abstention) that reduces UnF1, (2) prompt engineering fails to balance audacity (over-answering) and timidity (over-abstention), and (3) model scale correlates positively with UnF1 (with a sharp improvement at a critical parameter threshold). These findings inform both MLLM evaluation and optimization.
4. Practical distillation framework for open-source MLLMs. The proposed distillation strategies (UnD-QA, UnD-IQ, UnD-Joint) directly address the gap in uncertainty awareness between commercial (e.g., GPT-4o) and open-source models. Crucially, the paper identifies that distilling question generation (UnD-IQ) – not just question answering (UnD-QA) – drives the largest, most consistent UnF1 gains (12–17% vs. 2% from UnD-QA), providing a clear path to improving smaller, accessible MLLMs.

**Weaknesses:**

1. Underspecified algorithmic details for reproducibility.
2. The paper contrasts UnSAF with traditional metrics (Brier score, ECE) but omits direct comparisons to recent uncertainty-aware approaches for MLLMs, such as [1][2].
3. The paper identifies that instruction-tuned MLLMs are "more timid" (higher abstention, lower timidity scores) but does not investigate why this occurs. Is it due to reinforcement learning from human feedback (RLHF) prioritizing avoiding errors over answering?
4. While the paper claims distillation reduces hallucinations (via POPE and CHAIR), it does not define how hallucinations are linked to uncertainty awareness.

[1] Consistency and Uncertainty: Identifying Unreliable Responses From Black-Box Vision-Language Models for Selective Visual Question Answering, CVPR 2024.

[2] Exploring response uncertainty in mllms: An empirical evaluation under misleading scenarios, EMNLP 2025.

**Questions:**

1. Does a higher UnF1 score directly correlate with fewer object hallucinations (measured by POPE) or less semantic inconsistency (measured by CHAIR)?
2. the positive correlation between model scale and UnF1 is observed but not explained, e.g., do larger models have better spatial/linguistic understanding, or do they simply learn more "abstention triggers"?
3. if a model never abstains (FU=0, TU=0), how is timidity computed?

---

> ### Author Response · Authors · 2025-11-25
> **Responses to Reviewer 3F9m [1/3]**
>
> Thanks for your valuable time and comments. To address your concerns, we provide pointwise responses below.
> > _**W1.**_ Underspecified algorithmic details for reproducibility.
>
> **A1**
> We treat reproducibility with the utmost importance and would like to clarify the resources provided.
> As explicitly stated in the **Reproducibility Statement** of the main paper line 486, we have submitted the full anonymous source code as supplementary material to the OpenReview system. Beyond the code, we have provided an overview of UnSAF in **Figure 1** to visualize the pipeline, as well as all prompt templates used for generation and evaluation in **Appendix I**.
>
> We are fully committed to open-sourcing the code, model checkpoints, and datasets upon acceptance to ensure broad accessibility and reproducibility.
>
>
> > _**W2.**_ The paper contrasts UnSAF with traditional metrics (Brier score, ECE) but omits direct comparisons to recent uncertainty-aware approaches for MLLMs, such as [1][2].
>
> **A2.**
> We appreciate the constructive suggestion to compare against recent uncertainty-aware approaches for MLLMs. Following the reviewer's guidance, we reproduced and compared two metrics:
> * Misleading Rate (MR) [1]: measures the proportion of answers that flip from correct to incorrect under misleading prompts. Since MR is **applicable only to multi-choice tasks**, we evaluated it on five multi-choice datasets, consistent with these in [1].
> * Consistency and Uncertainty (CU) [2]: evaluates consistency between the model's answer to an original question and a rephrased version. Despite applying fuzzy matching strategies (e.g., ignoring case or normalizing number formats), the metric remains **susceptible to false mismatches**, undermining its reliability.
>
> We evaluated these metrics against the UnF1 score on two critical dimensions: **Rank Consistency** and **Sample Size Robustness**.
>
> (1) As shown in (https://anonymous.4open.science/r/ICLR-10CC/mr.png), MR and CU exhibit significant volatility in model rankings across different datasets, whereas UnF1 demonstrates remarkable consistency and stability. (2) As shown in (https://anonymous.4open.science/r/ICLR-10CC/sample_amount_sensitivity.png), while MR and CU are less sensitive to data size than ECE/Brier Score, they remain significantly more sensitive than the UnF1 score. They show large fluctuations when data is scarce, whereas the UnF1 score demonstrates superior stability even with fewer samples.
>
> These comparisons confirm that while MR and CU capture specific aspects of uncertainty, the **UnF1 score offers a more robust and stable metric** for benchmarking MLLMs.
>
> [1] Consistency and Uncertainty: Identifying Unreliable Responses From Black-Box Vision-Language Models for Selective Visual Question Answering, CVPR 2024.
>
> [2] Exploring response uncertainty in mllms: An empirical evaluation under misleading scenarios, EMNLP 2025.
>
> > _**W3.**_ The paper identifies that instruction-tuned MLLMs are "more timid" (higher abstention, lower timidity scores) but does not investigate why this occurs. Is it due to reinforcement learning from human feedback (RLHF) prioritizing avoiding errors over answering?
>
> **A3.**
> Regarding the mechanism of why instruction-tuned MLLMs are "more timid",recent literature [1, 2, 3] illustrate that instruction tuning significantly alters the model's decision landscape. Both [1] and [3] show that supervised fine-tuning can induce over conservative behaviors when tuning on datasets that explicitly contain abstention or refusal behaviors.
>
> Mechanistically, as highlighted in [2], instruction-tuned models exhibit a strong "certainty bias", consistently preferring certain outcomes (like refusal) to avoid potential errors. When faced with uncertain queries, this bias drives the model to adopt a safe, conservative strategy rather than risking a low-confidence answer. Our UnSAF experiments empirically validate this theoretical observation and the results show that instruction-tuned model seems to more timid.
>
> Regarding the reviewer's specific question on RLHF, we would like to clarify that the timidity observed in our experiments is primarily driven by SFT. However, the reviewer's intuition is insightful: while SFT initiates this behavior, RLHF would likely exacerbate it. Since reward models in RLHF typically penalize incorrect answers more heavily than abstention, this imposes a "safety tax" leading to even greater caution.
>
> [1] Know Your Limits: A Survey of Abstention in Large Language Models, TACL 2025
>
> [2] Instructed to Bias: Instruction-Tuned Language Models Exhibit Emergent Cognitive Bias, TACL 2024
>
> [3] The art of saying no: Contextual noncompliance in language models. NeurIPS 2024

---

> ### Author Response · Authors · 2025-11-25
> **Responses to Reviewer 3F9m [2/3]**
>
> > **L4&Q1.** While the paper claims distillation reduces hallucinations (via POPE and CHAIR), it does not define how hallucinations are linked to uncertainty awareness. Does a higher UnF1 score directly correlate with fewer object hallucinations (measured by POPE) or less semantic inconsistency (measured by CHAIR)?
>
> **A4.**
> The UnF1 score measures the model's ability to distinguish between what it knows and what it does not. Consequently, when the model encounters queries beyond its knowledge scope, the model with a high UnF1 score can correctly chooses to abstain rather than fabricating an answer.
>
> This mechanism directly impacts hallucination metrics. For POPE that measures object hallucination, when the model is uncertain about an object's existence, it chooses to abstain or answer negatively. Similarly, for CHAIR that measures semantic inconsistency, a model with a high UnF1 score avoids generating descriptions that exceed its visual evidence.
>
> To analyze this relationship empirically, we evaluated the UnF1 score of dedicated hallucination mitigation methods: DoLa [1] and CoVe [2]:
>
> | Model | Method | UnF1 $\uparrow$ | POPE F1 (%) $\uparrow$ | CHAIR_i $\downarrow$ | CHAIR_i F1 (%) $\uparrow$ |
> | :--- | :--- | :---: | :---: | :---: | :---: |
> | InternVL-8B | Baseline | 0.76 | 83.98 | 10.21 | 73.70 |
> | | DoLa | 0.60 | **87.70** | 10.02 | 73.88 |
> | | CoVe | 0.73 | 87.10 | 9.67 | 73.81 |
> | | UnD-QA | 0.70 | 86.01 | **9.60** | 73.40 |
> | | UnD-IQ | **0.90** | 83.63 | 10.89 | 74.00 |
> | | UnD-Joint | 0.83 | 87.11 | 9.72 | **74.10** |
> | Qwen-VL-Chat-7B | Baseline | 0.58 | 86.19 | 8.49 | 73.60 |
> | | DoLa | 0.59 | 79.00 | **8.48** | 77.43 |
> | | CoVe | 0.57 | 78.30 | 8.56 | 77.13 |
> | | UnD-QA | 0.72 | 84.19 | 9.87 | 74.50 |
> | | UnD-IQ | 0.72 | **90.68** | 9.50 | **77.90** |
> | | UnD-Joint | **0.79** | 90.43 | 9.90 | 77.00 |
> | mPLUG-Owl-7B | Baseline | 0.61 | 86.34 | 11.14 | 72.30 |
> | | DoLa | 0.60 | 79.11 | 10.02 | 73.88 |
> | | CoVe | 0.62 | 79.00 | 7.07 | 72.83 |
> | | UnD-QA | 0.55 | 91.36 | **6.95** | 75.10 |
> | | UnD-IQ | **0.75** | 88.01 | 10.31 | **78.30** |
> | | UnD-Joint | 0.73 | **91.60** | 9.08 | 76.50 |
>
> As shown in the table, while these methods can effectively reduce hallucinations, they do **not consistently improve the UnF1 score**. This suggests that they focus on correcting the final output via decoding interventions, without necessarily improving the model's internal uncertainty awareness. The results strongly support our claim that the UnF1 score captures a unique property, and hallucination mitigation methods do not consistently affect the UnF1 score.
>
> [1] DoLa: Decoding by Contrasting Layers Improves Factuality in Large Language Models, ICLR 2024.
>
> [2] Chain-of-verification reduces hallucination in large language models, ACL 2024.
>
>
> > _**Q2.**_ The positive correlation between model scale and UnF1 is observed but not explained, e.g., do larger models have better spatial/linguistic understanding, or do they simply learn more "abstention triggers"?
>
> **A5.**
> To investigate whether larger models simply learn more "abstention triggers" (i.e., becoming lazy or overly conservative), we analyzed the absolute counts of abstentions across model scales.
>
> | model | TU | FU | Number of Abstention |
> | :---|:------:|:------:|:-------:|
> | LLaVA-7B | 1645 | 1161 | 2806  |
> | LLaVA-13B | 1251 | 360 | 1611  |
> | Qwen2p5-VL-Instruct-7B | 3190 | 6142 | 9332  |
> | Qwen2p5-VL-Instruct-32B | 623 | 6094 | 6717  |
> | InternVL-14B | 747 | 8351 | 9098  |
> | InternVL-8B | 2656 | 3745 | 6401  |
> | mPLUG-Owl-2B | 2982 | 3978 | 6960  |
> | mPLUG-Owl-7B | 3682 | 5671 | 9353  |
> | MiniCPM-3B  | 4796 | 4924 | 9720  |
> | MiniCPM-8B  | 3178 | 5372 | 8550  |
>
> As shown in the table, **larger models do not simply abstain more**. For families like LLaVA, Qwen, and MiniCPM, the larger models actually exhibit fewer overall number of abstentions than their smaller counterparts. This observation aligns with established scaling laws in uncertainty literature, which indicate that larger models possess superior self-knowledge and calibration capabilities [1, 2]. Therefore, the performance improvement is attributed to enhanced spatial and linguistic understanding, allowing the model to correctly answer difficult queries rather than relying on heuristic abstention triggers.
>
> [1] Language Models (Mostly) Know What They Know, arXiv 2022.
>
> [2] Beyond the Imitation Game: Quantifying and extrapolating the capabilities of language models, TMLR 2023.

---

> ### Author Response · Authors · 2025-11-25
> **Responses to Reviewer 3F9m [3/3]**
>
> > _**Q3.**_ if a model never abstains (FU=0, TU=0), how is timidity computed?
>
> **A6.**
> In our framework, the refered metrics are normalized, such that **$TA + FU = 1$** and **$TU + FA = 1$**. If a model **never abstains ($FU=0, TU=0$)**, the outcome is as follows: Since $FU=0$, it follows that **$TA=1$**; Since $TU=0$, it follows that **$FA=1$**. Consequently, the metrics are computed as follows:
> * **Audacity** $= \frac{TA}{TA+FA} = \frac{1}{1+1} = \mathbf{0.5}$,
> **Timidity** $= \frac{TA}{TA+FU} = \frac{1}{1+0} = \mathbf{1.0}$,
> **UnF1** $= \frac{2 \cdot 0.5 \cdot 1.0}{0.5 + 1.0} = \mathbf{2/3} \approx \mathbf{0.67}$
>
>
> Furthermore, if a model **always abstains ($TA=0, FA=0$)**, then **Timidity** = **0**, which means that **Audacity**  is mathematically **undefined** (division by zero). In this specific edge case, we explicitly define the UnF1 score as **0**, reflecting a complete failure in uncertainty awareness. We thank the reviewer for this suggestion. We will add a more extensive discussion on this point in the final version of the paper.

---

### Official Review · Reviewer_CXVY · 2025-10-28

**Soundness:** 2
**Presentation:** 3
**Contribution:** 2
**Rating:** 4
**Confidence:** 3

**Summary:**

This paper introduces UnSAF (Uncertainty-Aware Self-Assessment Framework), a label-free evaluation framework for assessing uncertainty awareness in Multimodal Large Language Models (MLLMs). The core innovation is a two-stage pipeline where MLLMs first generate both answerable and unanswerable questions, then attempt to answer them. Responses are categorized into four types (True/False Answerable/Unanswerable), yielding an interpretable UnF1 score. The paper demonstrates that larger models exhibit better uncertainty awareness, motivates knowledge distillation to improve smaller MLLMs, and shows that uncertainty-aware question generation is critical for effective distillation.

**Strengths:**

1. Novel Problem Formulation: The paper addresses a genuinely important problem—uncertainty awareness in MLLMs—with a creative, self-contained approach that doesn't require extensive labeled data or external metrics. The UnF1 score formulation is intuitive and interpretable.

2. Comprehensive Empirical Study: The evaluation across 14 open-source and 3 commercial MLLMs is thorough. The consistency of findings regarding scaling effects and the effectiveness of UnD-Joint distillation strengthens the claims.

3. Principled Approach to Knowledge Distillation: The paper goes beyond simple QA-based distillation by incorporating uncertainty-aware question generation. The finding that this contributes 15% improvement (vs. 2% for QA-only) is significant and well-motivated.

**Weaknesses:**

1. Limited Theoretical Justification: While the UnF1 score is intuitive, the paper lacks deeper theoretical analysis of why this specific formulation effectively captures uncertainty. Why are the four categories optimal? How sensitive is the metric to the balance between answerable and unanswerable questions?

2. Validation Concerns:
   - The UnSAF validity analysis relies mainly on consistency across datasets and sampling stability, but lacks validation against ground-truth uncertainty measures
   - The paper doesn't clearly validate that UnF1 scores actually correlate with downstream metrics (e.g., error rates in real applications)

3. Dataset Annotation Quality: The distillation datasets are annotated using GPT-4o and peer MLLMs. How reliable are peer MLLM annotations? What's the inter-annotator agreement or quality assurance mechanism?

4. Limited Analysis of Failure Cases: The paper doesn't deeply discuss when UnSAF might fail or produce misleading results. Are there types of MLLMs or tasks where the approach breaks down?

**Questions:**

1. How sensitive is UnF1 to the ratio of answerable vs. unanswerable questions generated in Stage 1?

2. Can you provide correlation analysis between UnF1 scores and actual error rates across different model families?

3. For the distillation approach, how were the hyperparameters (LoRA rank, learning rate, etc.) selected?

4. The paper mentions instruction-tuned MLLMs adopt "timid behavior." Can you elaborate on mechanisms causing this and whether other types of tuning lead to different behaviors?

5. How does performance vary when applying UnSAF to models trained with different alignment procedures (e.g., DPO vs. RLHF)?

6. Scalability Questions:
   - How does computational cost scale with model size?
   - What's the overhead of the two-stage pipeline compared to single-pass methods?

---

> ### Author Response · Authors · 2025-11-25
> **Responses to Reviewer CXVY [1/3]**
>
> Thanks for your valuable time and comments. To address your concerns, we provide pointwise responses below.
>
> >_**W1&Q1.**_ Limited Theoretical Justification: While the UnF1 score is intuitive, the paper lacks deeper theoretical analysis of why this specific formulation effectively captures uncertainty. Why are the four categories optimal? How sensitive is the metric to the balance between answerable and unanswerable questions? How sensitive is UnF1 to the ratio of answerable vs. unanswerable questions generated in Stage 1?
>
> **Response.**
> The proposed formulation is theoretically grounded in the **confusion matrix** for binary classification. Specifically, the model's generation label in the UnQ-Gen stage is treated as the ground truth, while its answering behavior in the UnQ-Ans stage serves as the prediction. This mapping naturally yields four optimal categories. Structural analogues to Precision and Recall are defined as Audacity and Timidity. To effectively balance them, the harmonic mean is employed as **the UnF1 score**. This formulation guarantees that the metric captures robust uncertainty awareness, preventing the model from achieving high scores through trivial strategies.
>
> To investigate the **sensitivity to the ratio of answerable to unanswerable questions**, we vary the ratio of answerable to unanswerable questions ranging from 0.12 to 8.00. As shown in the table below, the UnF1 score demonstrates remarkable stability across extreme variations in data distribution especially for multi-choice setting. This confirms that the UnF1 score is a robust and distribution-agnostic metric.
>  | ratio | **InternVL-8B** |        | **LLaVA-13B** |        | **LLaVA-7B** |        | **Qwen-VL-Chat-7B** |        | **mPLUG-Owl-7B** |        |
> |:-----:|:---------------:|:------:|:-------------:|:------:|:------------:|:------:|:-----------:|:------:|:------------:|:------:|
> |       | MC              | OE     | MC            | OE     | MC           | OE     | MC          | OE     | MC           | OE     |
>  | 0.12  | 0.71            | 0.64   | 0.69          | 0.68   | 0.67         | 0.62   | 0.67        | 0.42   | 0.68         | 0.61   |
>  | 0.17  | 0.71            | 0.64   | 0.69          | 0.68   | 0.67         | 0.62   | 0.67        | 0.43   | 0.68         | 0.61   |
> | 0.25  | 0.71            | 0.64   | 0.69          | 0.69   | 0.67         | 0.62   | 0.67        | 0.43   | 0.68         | 0.61   |
>  | 0.50  | 0.72            | 0.65   | 0.69          | 0.68   | 0.67         | 0.62   | 0.67        | 0.42   | 0.68         | 0.60   |
>  | 0.67  | 0.71            | 0.64   | 0.69          | 0.67   | 0.67         | 0.67   | 0.67        | 0.43   | 0.68         | 0.61   |
>  | 2.00  | 0.70            | 0.63   | 0.69          | 0.69   | 0.67         | 0.69   | 0.67        | 0.45   | 0.68         | 0.62   |
> | 3.00  | 0.71            | 0.65   | 0.69          | 0.67   | 0.67         | 0.67   | 0.67        | 0.43   | 0.68         | 0.60   |
> | 8.00  | 0.71            | 0.65   | 0.69          | 0.68   | 0.67         | 0.68   | 0.67        | 0.44   | 0.68         | 0.60   |

---

> ### Author Response · Authors · 2025-11-25
> **Responses to Reviewer CXVY [2/3]**
>
> > _**W2&Q2.**_
> Validation Concerns: a) The UnSAF validity analysis relies mainly on consistency across datasets and sampling stability, but lacks validation against ground-truth uncertainty measures. b) The paper doesn't clearly validate that UnF1 scores actually correlate with downstream metrics (e.g., error rates in real applications). Can you provide correlation analysis between UnF1 scores and actual error rates across different model families?
>
> **Response.**
> Thanks for your suggestion. While we argue that ECE, often considered a ground-truth uncertainty measure, has **significant limitations** as a reliable uncertainty metric in open-ended VQA as discussed in Sections 3.1 and 4.1, we provide the requested validation to illustrate the relationship. As shown in the table below, there are times that the UnF1 score correlates with calibration. For instance, when ECE spikes to 0.43 on mPLUG-Owl-7B under UnD-QA, the UnF1 score drops significantly. However, an improved UnF1 score does not necessarily imply a lower ECE which confirms that **UnD-IQ improves the model's uncertainty awareness without compromising its token-level probability calibration**.
> | Model   | Method     |      GPT-4o| |      Moonshot-v1      | |     InternVL-14B      | |
> |-------------------|------------|:------------------:|:------:|:----------------------:|:------:|:----------------------:|:------:|
> | | | **UnF1**$\uparrow$ | **ECE**$\downarrow$ | **UnF1**$\uparrow$ | **ECE**$\downarrow$ | **UnF1**$\uparrow$ | **ECE**$\downarrow$ |
> | InternVL-8B   | base| 0.76 | 0.03 | 0.76 | 0.03 | 0.76 | 0.03 |
> | | UnD-QA     | 0.62 | 0.02 | 0.62 | 0.02 | 0.70 | 0.09 |
> | | UnD-IQ     | 0.91 | 0.02 | 0.91 | 0.02 | 0.90 | 0.02 |
> | | UnD-Joint  | 0.93 | 0.02 | 0.93 | 0.02 | 0.83 | 0.08 |
> | mPLUG-Owl-7B  | base| 0.61 | 0.06 | 0.61 | 0.06 | 0.61 | 0.06 |
> | | UnD-QA     | 0.52 | 0.43 | 0.61 | 0.09 | 0.55 | 0.12 |
> | | UnD-IQ     | 0.74 | 0.07 | 0.85 | 0.08 | 0.75 | 0.09 |
> | | UnD-Joint  | 0.95 | 0.07 | 0.92 | 0.07 | 0.73 | 0.12 |
> | Qwen-VL-Chat-7B | base     | 0.58 | 0.03 | 0.58 | 0.03 | 0.58 | 0.03 |
> | | UnD-QA     | 0.53 | 0.09 | 0.61 | 0.03 | 0.58 | 0.03 |
> | | UnD-IQ     | 0.70 | 0.06 | 0.68 | 0.07 | 0.72 | 0.03 |
> | | UnD-Joint  | 0.76 | 0.10 | 0.71 | 0.03 | 0.79 | 0.05 |
>
> Regarding the correlation with downstream metric like error rates, we provide empirical evidence from two perspectives.
> First, figure 6 of the main paper shows a clear positive correlation between the UnF1 score and accuracy that corresponds to a negative correlation with error rates.
>
> Furthermore, as shown in these two figures(https://anonymous.4open.science/r/ICLR-10CC/rejected_acc1.png, https://anonymous.4open.science/r/ICLR-10CC/rejected_acc2.png), distilled models demonstrate higher rejection accuracy across mPlug-Owl-7B and Qwen-VL-Chat-7B implying that a higher UnF1 score can be translated into a stronger ability to mitigate errors by rejecting uncertain queries in real-world applications.
>
>
> > _**W3.**_ Dataset Annotation Quality: The distillation datasets are annotated using GPT-4o and peer MLLMs. How reliable are peer MLLM annotations? What's the inter-annotator agreement or quality assurance mechanism?
>
> **Response.**
> Thanks for your question. It is important to clarify that the dataset that we intend to open-source is constructed using GPT-4o. The inclusion of peer MLLMs (e.g., Moonshot-v2, InternVL-14B) was not to replace GPT-4o as the primary data source, but specifically to demonstrate the effectiveness of our UnD-IQ strategy. Even if peer annotations contain slightly more noise than GPT-4o, UnD-IQ still consistently outperforms baselines, highlighting the method's robustness.
>
>
> >_**W4.**_ Limited Analysis of Failure Cases: The paper doesn't deeply discuss when UnSAF might fail or produce misleading results. Are there types of MLLMs or tasks where the approach breaks down?
>
> **Response.**
> We appreciate the suggestion to discuss failure cases. We have identified that **insufficient generation capability** is a critical failure mode. That is, if a model cannot follow instructions to properly generate questions, the UnSAF naturally fails.
>
> Besides, UnSAF relies on the model's answering behavior to calculate the UnF1 score. If a model **abstain from answering all self-generated questions**, resulting in TA = 0 and FA = 0, the audacity score becomes mathematically undefined (division by zero) and the timidity drops to zero. To address this, we explicitly define the UnF1 score as 0 in such cases. The case where timidity is undefined (TA=FU=0) does not occur, as TA+FU represents the total count of generated answerable questions, which is guaranteed to be non-zero given successful generation.

---

> ### Author Response · Authors · 2025-11-25
> **Responses to Reviewer CXVY [3/3]**
>
> > _**Q3.**_ For the distillation approach, how were the hyperparameters (LoRA rank, learning rate, etc.) selected?
>
> **Response.**
> We utilized the standard **default hyperparameters** recommended in the official Qwen-VL-Chat repository for LoRA fine-tuning. We deliberately maintained a **consistent hyperparameter setting** across all settings that ensures a fair comparison, aiming to isolate the effectiveness of the distillation strategy itself rather than attributing performance gains to specific hyperparameter tuning. The fact that UnD-IQ achieves superior performance under these standard settings further demonstrates its robustness and ease of use.
>
>
>
> >_**Q4.**_ The paper mentions instruction-tuned MLLMs adopt "timid behavior." Can you elaborate on mechanisms causing this and whether other types of tuning lead to different behaviors?
>
> **Response.**
> Thanks for your question. As discussed in recent literature [1, 2, 3], instruction tuning significantly alters the model's decision landscape.
> Both [1] and [3] show that supervised fine-tuning can induce over conservative behaviors when tuning on datasets that explicitly contain abstention or refusal behaviors.
>
> Mechanistically, as highlighted in [2], instruction-tuned models exhibit a strong **"certainty bias"**, consistently preferring certain outcomes (like refusal) to avoid potential errors. When faced with uncertain queries, this bias drives the model to adopt a safe, conservative strategy rather than risking a low-confidence answer. Our UnSAF experiments empirically validate this theoretical observation and the results show that instruction-tuned model seems to more timid.
>
> [1] Know Your Limits: A Survey of Abstention in Large Language Models, TACL 2025
>
> [2] Instructed to Bias: Instruction-Tuned Language Models Exhibit Emergent Cognitive Bias, TACL 2024
>
> [3] The art of saying no: Contextual noncompliance in language models. NeurIPS 2024
>
>
> > _**Q5.**_ How does performance vary when applying UnSAF to models trained with different alignment procedures (e.g., DPO vs. RLHF)?
>
> **Response.**
>
> In this work, we primarily observed that instruction tuning induces a 'timid' tendency compared to pre-trained models. We agree that applying the UnSAF to evaluate models trained with different alignment procedures is important and a lies in **integrating UnSAF into the alignment training loop itself**. While it requires a systematic redesign of the training pipeline, it remains beyond the scope of our paper.
>
> Furthermore, based on the mechanisms discussed in response to _Q4_, we can hypothesize that alignment techniques like RLHF/DPO would likely **exacerbate the "timid behavior."** Models that are imposed "safety tax" through alignment are expected to exhibit increased timidity in UnSAF.
>
> We have explicitly added a discussion on alignment procedures in the **Discussion and Limitations section**, marking it as a critical direction for future research.
>
>
> >**Q6.** Scalability Questions: a) How does computational cost scale with model size? b) What's the overhead of the two-stage pipeline compared to single-pass methods?
>
> **Response.**
> We conducted experiments on **NVIDIA H100 (80GB) GPUs** to evaluate the computational cost. The average time consumption is reported below:
> | Model          | UnQ-Gen(s) | UnQ-Ans(s) | Overall(s) |
> |----------------|:-------:|:-------:|:-------:|
> | LLaVA-7B       | 22.78   | 2.01    | 24.79   |
> | LLaVA-13B      | 24.67   | 2.99    | 27.66   |
> | InternVL-8B    | 22.30   | 2.23    | 24.53   |
> | InternVL-14B   | 28.17   | 4.01    | 32.18   |
> | mPLUG-Owl-2B   | 21.76   | 4.76    | 26.52   |
> | mPLUG-Owl-7B   | 22.40   | 4.69    | 27.09   |
>
> As shown in table, the computational cost demonstrates **efficient sub-linear scaling** with respect to model size. For example, scaling the LLaVA family from 7B to 13B (approx. 2x size) results in only an 11.6% increase in overall latency (24.79s vs. 27.66s). Similarly, scaling InternVL family from 8B to 14B results in a 31.2% increase.
>
> We can also observe that, the significant latency difference in two stages stems directly from the disparity in output token length. The UnQ-Gen stage generates a long sequence of multiple questions, whereas the UnQ-Ans stage typically involves short and concise responses.
>
> Moreover, our method exhibits high sample efficiency. Empirical results show that reliable and stable scores can be obtained with only a few hundred samples, which effectively limits the overall computational cost.

---

### Official Review · Reviewer_nAFs · 2025-10-29

**Soundness:** 2
**Presentation:** 3
**Contribution:** 2
**Rating:** 4
**Confidence:** 4

**Summary:**

The paper proposes UnSAF, a framework to evaluate the ability of MLLMs to provide self-assessment (without labelled data) of whether the models "know what they don't know" and correctly indicate whether a question is unanswerable or not. This is done via two stages: (1) the MLLM is prompted to generate answerable and unanswerable questions given images; (2) the MLLM is asked to answer the generated question and indicate whether a question is unanswerable. From these two stages, a confusion matrix of responses could be computed, and the precision/recall and F1 scores (audacity, timidity, and UnF1 respectively) are used as indication of the model's capability to abstain well. The authors showed that these scores show better consistency across datasets than other metrics, and observations such as UnSAF score trends with respect to model size and instruction-tuned models. The authors also developed a knowledge distillation framework for UnSAF (with QA pairs and Instruction-Question pairs), and results on how the distillation helps improve UnSF scores.

**Strengths:**

- The paper tackles an important problem of multimodal LLM reliability and uncertainty quantification.
- The content is well structured.
- Experiments are run on a wide range of MLLMs.
- The proposed perspective of using MLLMs to generate questions that it deem unanswerable and subsequently testing whether it can consistently assess if these questions are unanswerable seems novel.

**Weaknesses:**

- The paper seems to mix up various notions of uncertainty, leading to claims and experimental designs that may not be well substantiated. The proposed framework does not measure uncertainty of model responses in a similar way as Brier Score or ECE, and hence it is a conceptual mismatch for the authors to compare them. Brier Score and ECE evaluates probability calibration for each task query, and hence requires ground truth data for evaluation, while UnSAF evaluates the model's consistency in abstaining from responding to questions it cannot answer. The experimental design in Sec 4.1 may not be useful -- it might not be the case that model ranking has to be preserved for Brier Score and ECE across datasets, e.g. some models may be better trained and calibrated in some tasks or domains than others. The authors may want to consider repositioning their paper to clearly distinguish what UnSAF is evaluating compared to the various other metrics and uncertainty quantification related work.

- The proposed framework seems to specifically measure a given model's consistency, between question-generation of unanswerable questions and question-answering evaluation of unanswerable questions. This does not necessarily provide a clear indication of whether the proposed score will be useful independently in assessing a model's uncertainty in tackling various tasks. Unfortunately, most experimental results in the paper have been focused on consistency (including the distillation framework that is evaluated primarily only on the UnSAF metrics), without any other baselines or metrics. There is a mention of hallucination mitigation results at the end, but even then the results presented were relative (W/T/L counts) before and after distillation, rather than a more objective measure of the effectiveness of the proposed score.

- While the distillation strategy and results make sense when evaluated against specifically the UnSAF metrics (consistency as described above), it remains to be seen whether the approach yield better calibrated models from other objective metrics. There are some relative win-loss count results in Table 2 for hallucination mitigation, but reporting absolute metrics and comparisons with other hallucination mitigation methods would help strengthen the claims of the paper.

- The fidelity of the question generation process is unclear. Line 190 indicated that there seems to require manual verification of whether each set of options has exactly one unambiguous correct answer, raising questions on how the "ground truth" of generated questions are validated including for the open-ended questions. From the paper, it is unclear whether the framework expects that the model's assessment of what is answerable or unanswerable need to be correct from an objective perspective (as opposed to model's answerability), and whether it matters.

- The prompts used in the experiments seem relatively complex with specific rules. It would be useful to report how sensitive the entire UnF1 score is to variations in the prompts (e.g. even with just paraphrasing of the prompt). The prompt engineering experiments in Sec 4.2 seem to show relatively large sensitivity for open-ended questions, indicating that ablations on prompts may be useful for other parts of the paper.

- The paper positioned itself as related other multimodal LLMs, but the results seem to be restricted to just visual-language models. It would be useful for the authors to adjust the paper to be clearer about the scope and claims of the paper.

**Questions:**

Please see points of concern and questions in the weakness section.

---

> ### Author Response · Authors · 2025-11-25
> **Responses to Reviewer nAFs [1/3]**
>
> Thanks for your valuable time and comments. To address your concerns, we provide pointwise responses below.
> > _**Q1.**_
>  The paper seems to mix up various notions of uncertainty, leading to claims and experimental designs that may not be well substantiated.....  The authors may want to consider repositioning their paper to clearly distinguish what UnSAF is evaluating compared to the various other metrics and uncertainty quantification related work.
>
> **A1.**
> We thank the reviewer for pointing out the conceptual distinction.
> We agree that Brier Score and ECE measure probability calibration, whereas UnSAF evaluates answer and abstention consistency. _We have revised the paper to explicitly clarify this distinction._ Our intention in comparing them is not to equate them, but to demonstrate that traditional calibration metrics (ECE/Brier Score) are **ill-suited for open-ended** MLLM tasks, where defining "ground truth accuracy" is difficult in both token-level and sentence level.
>
> Regarding the experimental design, our hypothesis is that uncertainty awareness should ideally be an **intrinsic capability of the model**, rather than fluctuating wildly across datasets. The observed inconsistency in ECE/Brier Score rankings likely stems from their sensitivity to answer correctness definitions in open-ended generation, making them unstable metrics for benchmarking MLLMs. In contrast, UnSAF focuses on the model's self-consistency in recognizing knows and unknowns, which proves to be a more robust metric for evaluating the model's uncertainty awareness across domains. Additionally, we add direct comparisons to recent uncertainty-aware approaches, in our response to **Reviewer 3F9m (Question 2)**.
>
>
>
> > _**Q2.**_
> The proposed framework seems to specifically measure a given model's consistency, between question-generation of unanswerable questions and question-answering evaluation of unanswerable questions. This does not necessarily provide a clear indication of whether the proposed score will be useful independently in assessing a model's uncertainty in tackling various tasks. Unfortunately, most experimental results in the paper have been focused on consistency (including the distillation framework that is evaluated primarily only on the UnSAF metrics), without any other baselines or metrics......
>
> **A2.**
> By definition, the UnF1 score is an **intrinsic indicator of the model's uncertainty awareness**. A low UnF1 score indicates the model's inability to distinguish what it knows from what it does not, implying it cannot be trusted to perform safely in downstream tasks. Thus, the UnF1 score can dependently serves as a vital proxy for the model's reliability and trustworthiness. Crucially, our experiments reveal that most MLLMs currently exhibit low uncertainty awareness and suffer from distinct biases.
>
> While we argue that **ECE is unreliable for open-ended generation** (as discussed in _Q1_), we provide the ECE results on VQAv2 below as requested to further validate our method. As shown in the table, ECE exhibits minor and inconsistent fluctuations across models, whereas UnD-IQ consistently improves the UnF1 score. Crucially, UnD-IQ achieves significant gains in the UnF1 score while maintaining low ECE.
>
> | Model   | Method     |      GPT-4o| |      Moonshot-v1      | |     InternVL-14B      | |
> |-------------------|------------|:------------------:|:------:|:----------------------:|:------:|:----------------------:|:------:|
> | | | **UnF1**$\uparrow$ | **ECE**$\downarrow$ | **UnF1**$\uparrow$ | **ECE**$\downarrow$ | **UnF1**$\uparrow$ | **ECE**$\downarrow$ |
> | InternVL-8B   | Baseline| 0.76 | 0.03 | 0.76 | 0.03 | 0.76 | 0.03 |
> | | UnD-QA     | 0.62 | 0.02 | 0.62 | 0.02 | 0.70 | 0.09 |
> | | UnD-IQ     | 0.91 | 0.02 | 0.91 | 0.02 | 0.90 | 0.02 |
> | | UnD-Joint  | 0.93 | 0.02 | 0.93 | 0.02 | 0.83 | 0.08 |
> | mPLUG-Owl-7B  | Baseline| 0.61 | 0.06 | 0.61 | 0.06 | 0.61 | 0.06 |
> | | UnD-QA     | 0.52 | 0.43 | 0.61 | 0.09 | 0.55 | 0.12 |
> | | UnD-IQ     | 0.74 | 0.07 | 0.85 | 0.08 | 0.75 | 0.09 |
> | | UnD-Joint  | 0.95 | 0.07 | 0.92 | 0.07 | 0.73 | 0.12 |
> | Qwen-VL-Chat-7B | Baseline| 0.58 | 0.03 | 0.58 | 0.03 | 0.58 | 0.03 |
> | | UnD-QA     | 0.53 | 0.09 | 0.61 | 0.03 | 0.58 | 0.03 |
> | | UnD-IQ     | 0.70 | 0.06 | 0.68 | 0.07 | 0.72 | 0.03 |
> | | UnD-Joint  | 0.76 | 0.10 | 0.71 | 0.03 | 0.79 | 0.05 |
>
> As shown in the table above, the inconsistent fluctuations of ECE across models highlight its instability, whereas the UnF1 score shows consistent improvement with UnD-IQ. Crucially, UnD-IQ achieves significant gains in the UnF1 score while maintaining low ECE.
>
> To address the concern about the lack of comparisons with other methods, we have added comparisons with UNK-VQA and MM-UPD. Please refer to our response to **Reviewer zwRv (Question 3)** for the detailed results and analysis.

---

> ### Author Response · Authors · 2025-11-25
> **Responses to Reviewer nAFs [2/3]**
>
> > _**Q3.**_
> There is a mention of hallucination mitigation results at the end, but even then the results presented were relative (W/T/L counts) before and after distillation, rather than a more objective measure of the effectiveness of the proposed score...... While the distillation strategy and results make sense when evaluated against specifically the UnSAF metrics (consistency as described above), it remains to be seen whether the approach yield better calibrated models from other objective metrics. There are some relative win-loss count results in Table 2 for hallucination mitigation, but reporting absolute metrics and comparisons with other hallucination mitigation methods would help strengthen the claims of the paper.
>
> **A3.**
> Thanks for your comment. We have conducted supplementary experiments to directly address this concern. We compared our method with two hallucination mitigation baselines: DoLa [1] and CoVe [2]. The comparative results (POPE and CHAIR) are reported below:
>
> | Model | Method | POPE F1 (%) $\uparrow$ | CHAIR_i $\downarrow$ | CHAIR_i F1 (%) $\uparrow$ |
> | :--- | :--- | :---: | :---: | :---: |
> | InternVL-8B | Baseline | 83.98 | 10.21 | 73.70 |
> | | DoLa | **87.70** | 10.02 | 73.88 |
> | | CoVe | 87.10 | 9.67 | 73.81 |
> | | UnD-QA | 86.01 | **9.60** | 73.40 |
> | | UnD-IQ | 83.63 | 10.89 | 74.00 |
> | | UnD-Joint| 87.11 | 9.72 | **74.10** |
> | Qwen-VL-Chat-7B | Baseline | 86.19 | 8.49 | 73.60 |
> | | DoLa | 79.00 | **8.48** | 77.43 |
> | | CoVe | 78.30 | 8.56 | 77.13 |
> | | UnD-QA | 84.19 | 9.87 | 74.50 |
> | | UnD-IQ | **90.68** | 9.50 | **77.90** |
> | | UnD-Joint | 90.43 | 9.90 | 77.00 |
> | mPLUG-Owl-7B | Baseline | 86.34 | 11.14 | 72.30 |
> | | DoLa | 79.11 | 10.02 | 73.88 |
> | | CoVe | 79.00 | 7.07 | 72.83 |
> | | UnD-QA | 91.36 | **6.95** | 75.10 |
> | | UnD-IQ | 88.01 | 10.31 | **78.30** |
> | | UnD-Joint | **91.60** | 9.08 | 76.50 |
>
> As is shown, our distillation strategy significantly reduces hallucinations. It is worth noting that while DoLa and CoVe require complex inference-time interventions, our method achieves comparable or superior results via standard inference, making it more efficient for deployment. Furthermore, complex inference-time interventions may harm question-answering ability, as POPE scores are lower than the baseline, while CHAIR scores are not.
>
> [1] DoLa: Decoding by Contrasting Layers Improves Factuality in Large Language Models, ICLR 2024
>
> [2] Chain-of-Verification Reduces Hallucination in Large Language Models, ACL 2024
>
>
> > _**Q4.**_ The fidelity of the question generation process is unclear. Line 190 indicated that there seems to require manual verification of whether each set of options has exactly one unambiguous correct answer, raising questions on how the "ground truth" of generated questions are validated including for the open-ended questions. From the paper, it is unclear whether the framework expects that the model's assessment of what is answerable or unanswerable need to be correct from an objective perspective (as opposed to model's answerability), and whether it matters.
>
> **A4.**
> Thanks for your suggestion. The mention of "manual verification" refers strictly to a check on the **structural format** to ensure the presence of four options (i.e., avoiding cases with missing or too many options), rather than an assessment of semantic correctness. The phrase "contain exactly one unambiguously correct answer" describes the instruction requirements given to the model, not a criterion that was manually enforced on the output.
>
> Furthermore, the framework does not require the model's assessment of what is answerable or unanswerable to be correct from an objective perspective, as **the ability to generate specific type of questions** is essentially part of the uncertainty awareness assessment. Therefore, as implied by the reviewer, objective correctness is not the primary concern here.

---

> ### Author Response · Authors · 2025-11-25
> **Responses to RevieWer nAFs [3/3]**
>
> > _**Q5.**_ The prompts used in the experiments seem relatively complex with specific rules. It would be useful to report how sensitive the entire UnF1 score is to variations in the prompts (e.g. even with just paraphrasing of the prompt). The prompt engineering experiments in Sec 4.2 seem to show relatively large sensitivity for open-ended questions, indicating that ablations on prompts may be useful for other parts of the paper.
>
> **A5.**
> We appreciate the insightful suggestion to assess prompt sensitivity. To address this concern, we have employed GPT-5 to generate diverse semantic paraphrases of the original prompts and re-evaluated the models.
> Here, Para-1, Para-2, and Para-3 refer to three distinct paraphrased versions of the original prompt:
>
> | Model   || |    MC     | | \| |      | |  OE| |
> |-------------------|-----------------------|--------|--------|--------|--------|----------------|--------|--------|--------|
> | | original   | Para-1 |  Para-2 |  Para-3  |\|| original |  Para-1 |  Para-2 | Para-3 |
> | LLaVA-7B| 0.67 | 0.67 | 0.67 | 0.67 | \|| 0.65 | 0.67 | 0.67 | 0.67 |
> | Qwen-VL-Chat-7B    | 0.63 | 0.67 | 0.65 | 0.57 |  \||0.58 | 0.47 | 0.47 | 0.45 |
> | Moonshot-v1| 0.76 | 0.79 | 0.81 | 0.78 | \|| 0.89 | 0.89 | 0.89 | 0.89 |
> | GPT-4o   | 0.93 | 0.95 | 0.94 | 0.96 | \|| 0.84 | 0.84 | 0.84 | 0.84 |
>
> As shown in the table above, the **UnF1 score exhibits high stability** across prompt variations for the majority of models. This indicates that our UnF1 score is generally robust to prompt variations.
>
> Regarding the sensitivity of the UnF1 score observed in Section 4.2, we clarify that those experiments involved **leading prompts** which intentionally introduce bias. In contrast, standard paraphrasing as done here maintains the core instructional content, resulting in the observed stability.
>
> > _**Q6.**_ The paper positioned itself as related other multimodal LLMs, but the results seem to be restricted to just visual-language models. It would be useful for the authors to adjust the paper to be clearer about the scope and claims of the paper.
>
> **A6.**
> We acknowledge that our experiments primarily focus on vision-language modalities. The term "MLLMs" we adopted follows the prevalent consensus in recent literature, where it is standardly used to describe large models capable of processing visual and textual information:
> * Exploring response uncertainty in mllms: An empirical evaluation under misleading scenarios, EMNLP 2025
> * MLLMs Know Where to Look: Training-free Perception of Small Visual Details with Multimodal LLMs, ICLR 2025
> * UNK-VQA: A Dataset and a Probe Into the Abstention Ability of Multi-Modal Large Models, TPAMI 2024
>
> However, we agree that precision is important. In the revised version, we will explicitly define our scope at the beginning of the introduction, clarifying that in the context of this work, 'MLLM' specifically refers to Large Vision-Language Models.

---

### Official Review · Reviewer_zwRv · 2025-11-02

**Soundness:** 3
**Presentation:** 3
**Contribution:** 3
**Rating:** 4
**Confidence:** 3

**Summary:**

This paper addresses the challenge of reliably evaluating uncertainty in MLLMs, as existing metrics often require extensive labeled data and perform poorly on open-ended tasks. The authors introduce the UnSAF, a novel, label-free evaluation method.  Experiments across 17 MLLMs show that the UnF1 score is more stable and consistent than traditional metrics. The research also reveals a positive correlation between model scale and uncertainty awareness. Based on this finding, the authors propose an uncertainty-aware distillation method. They demonstrate that teaching a smaller model to generate uncertainty-aware questions is more effective at improving its UnF1 score and reducing hallucinations than simply fine-tuning on question-answer pairs, without degrading task performance.

**Strengths:**

1. The proposed UnSAF framework is a methodologically novel, label-free approach. It uniquely uses the model's own generated answerable and unanswerable questions to create an internal evaluation benchmark.
2. The paper rigorously validates its UnF1 metric. Figures 2 and 3 demonstrate superior stability across datasets and sample sizes compared to conventional metrics like ECE.

**Weaknesses:**

1.  The method's core relies on the model's ability to self-generate answerable and unanswerable questions, yet the experimental section lacks a human evaluation of the quality of these questions. It is recommended to add a human assessment component to validate the fundamental premise of the UnSAF framework's effectiveness.
2.  The uncertainty-aware distillation experiments only validated the UnF1 score improvement on the COCO test set, which shares the same distribution as the training data. It is suggested to evaluate the distilled models' UnF1 scores on more diverse, unseen datasets (e.g., MMBench, OKVQA) to demonstrate the generalizability of the enhanced uncertainty awareness.
3.  While the paper proposes an effective UnD distillation strategy, it lacks direct experimental comparisons with other recent methods aimed at improving model uncertainty or abstention capabilities (such as those mentioned in Miyai et al., 2024). Adding such baseline comparisons is recommended to more comprehensively demonstrate the advantages of the proposed method.
4.  The experimental results show that the UnD-QA strategy can lead to a decrease in the UnF1 score in some cases, but the authors fail to provide an in-depth analysis of this negative result. It is advisable to conduct a more detailed investigation into this phenomenon to explain why distilling only question-answer pairs might harm the model's uncertainty awareness.

**Questions:**

None

---

> ### Author Response · Authors · 2025-11-25
> **Response to Review zwRv[1/2]**
>
> Thanks for your valuable time and comments. To address your concerns, we provide pointwise responses below.
> > **_Q1._** The method's core relies on the model's ability to self-generate answerable and unanswerable questions, yet the experimental section lacks a human evaluation of the quality of these questions. It is recommended to add a human assessment component to validate the fundamental premise of the UnSAF framework's effectiveness.
>
> **A1.**
> hank you for your question. We agree that the quality of the questions is crucial for evaluating a model’s ability to handle uncertainty. However, it is also important to emphasize that the model’s capacity to generate high-quality questions is itself a key aspect of the evaluation. UnSAF is designed to evaluate the model as a complete system and consequently even if some models produce low-quality questions (e.g., ambiguous or ill-formed ones), these outputs reflect its genuine internal state and should be evaluated as such.
>
> Crucially, introducing human evaluation to manually filter out these questions would create an **artificial setting**, decoupling the generation capability from the answering capability. If we filter out "bad" questions, we are testing a processed version of the model, not the model itself.
>
> > _**Q2.**_ The uncertainty-aware distillation experiments only validated the UnF1 score improvement on the COCO test set, which shares the same distribution as the training data. It is suggested to evaluate the distilled models' UnF1 scores on more diverse, unseen datasets (e.g., MMBench, OKVQA) to demonstrate the generalizability of the enhanced uncertainty awareness.
>
> **A2.**
> We agree that evaluating on diverse, unseen datasets is crucial and consequently we conducted supplementary experiments on **MMBench and OKVQA**. The following table highlights the consistent and significant improvements:
>
> | model | strategy  | **GPT-4o** | | |      **Moonshot-v1**      | | |     **InternVL-14B**     | | |
> |:----------------:|:---------:|:------------------------:|:------:|:------:|:--------------------------:|:------:|:------:|:-------------------------:|:------:|:------:|
> ||| CoCo | MMBench| OKVQA  | CoCo | MMBench| OKVQA  | CoCo| MMBench| OKVQA  |
> | InternVL-8B  | Baseline      | 0.76 | 0.67   | 0.67   | 0.76 | 0.67   | 0.67   | 0.76| 0.67   | 0.67   |
> || UnD-QA    | 0.62 | 0.61   | 0.61   | 0.68 | 0.71   | 0.71   | 0.70| 0.75   | 0.77   |
> || UnD-IQ    | 0.91 | **0.89** | **0.92** | **0.96**      | **0.95** | **0.95** | **0.90**    | **0.82** | **0.81** |
> || UnD-Joint | **0.93**      | 0.83   | 0.87   | 0.95 | 0.94   | 0.94   | 0.83| 0.72   | 0.71   |
> | mPLUG-Owl-7B | Baseline      | 0.61 | 0.61   | 0.61   | 0.61 | 0.61   | 0.61   | 0.61| 0.61   | 0.61   |
> || UnD-QA    | 0.52 | 0.58   | 0.50   | 0.61 | 0.64   | 0.65   | 0.55| 0.59   | 0.56   |
> || UnD-IQ    | 0.74 | **0.94** | **0.96** | 0.85 | 0.82   | 0.85   | **0.75**      | **0.73** | **0.75** |
> || UnD-Joint | **0.95**      | **0.94** | 0.95   | **0.92**| **0.90** | **0.90** | 0.73 | 0.72   | 0.74   |
>
> As the table shown, UnD-IQ consistently outperforms UnD-QA on these datasets, with a maximum UnF1 score increase of 0.46 (from 0.50 to 0.96) and an average improvement of 0.24 across all setups. This confirms that the uncertainty awareness learned via UnD-IQ is robust and effectively generalizes beyond the training distribution. Consistent with the results reported in the main text, UnD-Joint also delivers highly stable performance across various settings.

---

> ### Author Response · Authors · 2025-11-25
> **Response to Review zwRv[2/2]**
>
> >> _**Q3.**_ While the paper proposes an effective UnD distillation strategy, it lacks direct experimental comparisons with other recent methods aimed at improving model uncertainty or abstention capabilities (such as those mentioned in Miyai et al., 2024). Adding such baseline comparisons is recommended to more comprehensively demonstrate the advantages of the proposed method.
>
> **A3.**
> Thanks for your suggestion. We selected two representative fine-tuning methods using **UNK-VQA [1]** and **MM-UPD [2]** for comparison. We chose these because:
> * **UNK-VQA** is specifically designed to address the challenge of identifying questions that models do not know. This method includes two variants: UNK-VQA (BY), which operates in a binary setting by predicting whether a question is "answerable" or "unanswerable"; UNK-VQA (OE), which functions in an open-ended setting by providing the actual answer when available.
> * **MM-UPD** is designed to examine the MLLM’s ability to withhold answers when faced with unsolvable problems.
>
> The comparative results (using GPT-4o as the teacher for our method) are presented below.
> | Model ||   InternVL-8B     | |     |  Qwen-VL-Chat-7B    | |      |mPLUG-Owl-7B    | |
> |-----------------|:-------------------------:|:------:|:------:|:----------------------:|:------:|:------:|:---------------------:|:------:|:------:|
> | **Method**     | A   |   T    |  UnF1  |A |   T    |  UnF1  | A |   T    |  UnF1  |
> | Baseline [2]     |0.65 |  0.90  |  0.76  |0.51|  0.67  |  0.58  | 0.59|  0.63  |  0.61  |
> | MM-UPD [2]     |0.65 |  0.77  |  0.71  |0.66|  0.55  |  0.60  | 0.55|  0.66  |  0.60  |
> | UNK-VQA (BY) [1] |/ |  /  |  /  |0.00|  0.00  |  0.00  | 0.76|  0.39  |  0.52  |
> | UNK-VQA (OE) [1] |0.53 |  0.98  |  0.69  |/|  /  |  /  | 0.53|  0.53  |  0.53  |
> | UnD-QA      |0.60 |  0.64  |  0.62  |0.58|  0.49  |  0.53  | 0.63|  0.45  |  0.52  |
> | UnD-IQ      |0.90 |  0.92  |  0.91  |0.70|  **0.70**  |  0.70  | 0.88|  0.64  |  0.74  |
> | UnD-Joint   |**0.91** |  **0.96**  |  **0.93**  |**0.88**|  0.67  |  **0.76**  | **0.93**|  **0.97**  |  **0.95**  |
>
> The results show that UNK-VQA and MM-UPD that rely solely on QA-pair fine-tuning demonstrate limited effectiveness. In sharp contrast, UnD-IQ achieves a superior uncertainty awareness, enabling the model to effectively decide when to answer and when to abstain.
>
> Regarding the failure cases in table, we observed two distinct issues. For UNK-VQA (OE) on Qwen-VL-Chat-7B and UNK-VQA (BY) on InternVL-8B, the tuned models partially lost the ability to generate valid questions. Meanwhile, the zero UnF1 score observed with UNK-VQA (BY) on Qwen-VL-Chat-7B indicate that the model defaulted to unanswerable outputs for nearly all self-generated questions.
>
> [1] UNK-VQA: A Dataset and a Probe into the Abstention Ability of Multi-modal Large Models, TPAMI 2024
>
> [2] Unsolvable Problem Detection: Robust Understanding Evaluation for Large Multimodal Models, ACL 2025
>
>
> > _**Q4**_. The experimental results show that the UnD-QA strategy can lead to a decrease in the UnF1 score in some cases, but the authors fail to provide an in-depth analysis of this negative result. It is advisable to conduct a more detailed investigation into this phenomenon to explain why distilling only question-answer pairs might harm the model's uncertainty awareness.
>
> **A4.**
> We appreciate the opportunity to elaborate on this phenomenon. The negative impact of UnD-QA observed in some cases is actually consistent with our findings in Section 4.2 regarding the lower $T$ of supervised finetuning. This negative result serves as crucial evidence supporting our motivation: merely enhancing the model's QA capabilities is insufficient for uncertainty awareness, highlighting the necessity of the proposed UnD-IQ strategy.
>
> Additionally, we provide a more in-depth analysis of the underlying mechanism, supported by related literature, in our response to **Reviewer 3F9m (Question 3)**.

---

### Author Response · Authors · 2025-12-01
**Official Comment [1/2]**

We thank the reviewers for their constructive feedback and are encouraged that they recognized several key strengths of our work:
* Importance and Novelty: Reviewers mentioned the problem we addressed is an **important** (nAFs) and even highlighted as **genuinely important** for multimodal LLM reliability and uncertainty awareness (CXVY). The UnSAF framework is recognized as methodologically **novel** (zwRv, CXVY) and **addressing a critical cap** in MLLM uncertainty assessment (3F9m).


* Interpretable and Effective Method: Reviewers highlighted that the UnF1 score is **intuitive, interpretable** (CXVY), and demonstrates superior **stability** across datasets and sample sizes compared to conventional metrics such as ECE and MCE which makes UnSAF practical for low-data regimes (zwRv, 3F9m).

* Comprehensive Experiments: The evaluation was commended as **comprehensive** (nAFs, CXVY) **with insightful empirical observations** (3F9m). Additionally, the distillation strategy with instruction-question pair was viewed as **significant and well-motivated, providing a clear path to improving smaller, accessible MLLMs** (CXVY, 3F9m).


We appreciate above positive feedback and would like to address several common concerns below.

### (Concern 1) Quality of self-generated questions
We understand reviewer (zwRv) concern regarding the quality of the self-generated questions. We agree that the quality of the questions is crucial for evaluating a model's ability to handle uncertainty. However, it is also important to emphasize that the **model's capacity to generate high-quality questions itself is a key aspect of the evaluation**. UnSAF is designed to evaluate the model as a complete system and consequently even if some models produce low-quality questions (e.g., ambiguous or ill-formed ones), these outputs reflect its genuine internal state and should be evaluated as such.

Crucially, introducing human evaluation to manually filter out these questions would create an artificial setting, decoupling the generation capability from the answering capability. If we filter out "bad" questions, we are testing a processed version of the model, not the model itself.


### (Concern 2) Metric Validity
We thank the reviewers for raising questions on the lack of the comparision between the UnF1 score and other utility metrics (e.g., ECE) (Reviewer nAFs, CXVY).

While we argue that **ECE is unreliable for open-ended generation**, we provide the ECE results on VQAv2 below as requested to further validate our method.

| Model   | Method     |      GPT-4o| |      Moonshot-v1      | |     InternVL-14B      | |
|-------------------|------------|:------------------:|:------:|:----------------------:|:------:|:----------------------:|:------:|
| | | UnF1$\uparrow$ | ECE$\downarrow$ | UnF1$\uparrow$ | ECE$\downarrow$ | UnF1$\uparrow$ | ECE$\downarrow$ |
| InternVL-8B   | Baseline | 0.76 | 0.03 | 0.76 | 0.03 | 0.76 | 0.03 |
| | UnD-QA     | 0.62 | 0.02 | 0.62 | 0.02 | 0.70 | 0.09 |
| | UnD-IQ     | 0.91 | 0.02 | 0.91 | 0.02 | 0.90 | 0.02 |
| | UnD-Joint  | 0.93 | 0.02 | 0.93 | 0.02 | 0.83 | 0.08 |
| mPLUG-Owl-7B  | Baseline | 0.61 | 0.06 | 0.61 | 0.06 | 0.61 | 0.06 |
| | UnD-QA     | 0.52 | 0.43 | 0.61 | 0.09 | 0.55 | 0.12 |
| | UnD-IQ     | 0.74 | 0.07 | 0.85 | 0.08 | 0.75 | 0.09 |
| | UnD-Joint  | 0.95 | 0.07 | 0.92 | 0.07 | 0.73 | 0.12 |
| Qwen-VL-Chat-7B | Baseline     | 0.58 | 0.03 | 0.58 | 0.03 | 0.58 | 0.03 |
| | UnD-QA     | 0.53 | 0.09 | 0.61 | 0.03 | 0.58 | 0.03 |
| | UnD-IQ     | 0.70 | 0.06 | 0.68 | 0.07 | 0.72 | 0.03 |
| | UnD-Joint  | 0.76 | 0.10 | 0.71 | 0.03 | 0.79 | 0.05 |

As shown in the table above, **the inconsistent fluctuations of ECE across models highlight its instability, whereas the UnF1 score shows consistent improvement with UnD-IQ**. Crucially, UnD-IQ and UnD-Joint achieves significant gains in the UnF1 score while maintaining low ECE.

---

> ### Author Response · Authors · 2025-12-01
> **Official Comment [2/2]**
>
> ### (Concern 3) Missing Baselines
> We thank the reviewers for raising questions on the **lack of direct comparisons aimed at improving model uncertainty or abstention capabilities**.(zwRv, nAFs).
>
> We have now implemented the suggested UNK-VQA and MM-UPD as comparative baselines:
> | Model ||   InternVL-8B     | |     |  Qwen-VL-Chat-7B    | |      |mPLUG-Owl-7B    | |
> |-----------------|:-------------------------:|:------:|:------:|:----------------------:|:------:|:------:|:---------------------:|:------:|:------:|
> | **Method**     | A   |   T    |  UnF1  |A |   T    |  UnF1  | A |   T    |  UnF1  |
> | Baseline      |0.65 |  0.90  |  0.76  |0.51|  0.67  |  0.58  | 0.59|  0.63  |  0.61  |
> | MM-UPD [2]     |0.65 |  0.77  |  0.71  |0.66|  0.55  |  0.60  | 0.55|  0.66  |  0.60  |
> | UNK-VQA (BY) [1] |/ |  /  |  /  |0.00|  0.00  |  0.00  | 0.76|  0.39  |  0.52  |
> | UNK-VQA (OE) [1] |0.53 |  0.98  |  0.69  |/|  /  |  /  | 0.53|  0.53  |  0.53  |
> | UnD-QA      |0.60 |  0.64  |  0.62  |0.58|  0.49  |  0.53  | 0.63|  0.45  |  0.52  |
> | UnD-IQ      |0.90 |  0.92  |  0.91  |0.70|  **0.70**  |  0.70  | 0.88|  0.64  |  0.74  |
> | UnD-Joint   |**0.91** |  **0.96**  |  **0.93**  |**0.88**|  0.67  |  **0.76**  | **0.93**|  **0.97**  |  **0.95**  |
>
> The results show that UNK-VQA and MM-UPD that **rely solely on QA-pair fine-tuning demonstrate limited effectiveness**. In sharp contrast, **UnD-IQ and UnD-Joint achieve a superior uncertainty awareness**, enabling the model to effectively decide when to answer and when to abstain. _Please refer to line 507 of the main paper for a detailed explanation of the failure cases shown in above table._
>
> [1] UNK-VQA: A Dataset and a Probe into the Abstention Ability of Multi-modal Large Models, TPAMI 2024
>
> [2] Unsolvable Problem Detection: Robust Understanding Evaluation for Large Multimodal Models, ACL 2025

---

### Meta-Review · Area_Chair_LLER · 2026-01-07

**Summary:**

This paper proposes UnSAF, a framework for evaluating uncertainty awareness in multimodal large language models without relying on human-labeled uncertainty annotations. The approach measures whether a model can consistently distinguish answerable from unanswerable questions by generating such questions itself and then answering them, producing an UnF1 score.
Several reviewers appreciated the attempt to move beyond calibration metrics and focus on abstention behavior. At the same time, concerns were raised about conceptual clarity (what kind of uncertainty UnSAF captures), the reliance on self-generated questions without human validation, limited comparisons to existing uncertainty or abstention methods, and whether UnF1 meaningfully correlates with downstream reliability improvements, such as reduced hallucination. Additional concerns included generalization of the distillation results, prompt sensitivity, and the overall scope of the evaluation.

**Reviewer Concerns:**

### Concerns addressed by the rebuttal:

- The authors clarified that UnSAF is intended to measure uncertainty awareness and abstention consistency, rather than probabilistic calibration, and revised the paper to better distinguish UnF1 from metrics such as ECE.

- Additional experiments on datasets not used for training (e.g., MMBench and OKVQA) to show the generalization.

- Comparisons with existing methods such as UNK-VQA and MM-UPD were added, addressing concerns about missing baselines.

- New evaluations using POPE and CHAIR provided evidence that the proposed approach reduces hallucinations without requiring inference-time interventions.

- Prompt sensitivity analyses and clarifications.

- Computational overhead analysis.

### Concerns that remain outstanding:

- Quality of self-generated questions may fall into a circular reasoning difficulty.

- The theoretical grounding of UnF1 remains relatively light, beyond its confusion-matrix interpretation.

- The study focuses on vision–language models and extension to other modalities is not demonstrated.

- Failure cases are discussed but not analyzed in depth.

**Reviewer Scores:**

Reviewer zwRv: Initially 4. Given the added baselines and generalization experiments and the score is likely increase.

Reviewer nAFs: Initially 4. Most conceptual and empirical concerns were addressed and the score is likely unchanged.

Reviewer CXVY: Initially 4. The added analyses and clarifications would likely raise the score.

Reviewer 3F9m: Initially 4. Concerns were partially addressed, the score is likely unchanged.

---

### Decision · Program_Chairs · 2026-01-26

Reject